# Microtubules in the axon are GDP bound but adopt a stable GTP-like expanded state

Elena A. Zehr[1], Shufeng Sun[1], Stephanie L. Sarbanes[1] & Antonina Roll-Mecak ®[1,2] ✉

Microtubules scaffold cells, supporting signaling and cargo transport. They assemble from GTP–tubulin, which hydrolyzes to GDP–tubulin during polymerization. GTP–microtubule lattices are stable; GDP lattices depolymerize rapidly. In vitro, hydrolysis triggers lattice compaction. Lattice spacing regulates motors and microtubule-associated proteins; however, the conformation of tubulin in microtubules in cells is unknown. Here, we present the atomic-resolution cryo-electron microscopy structure of human microtubules in situ, in the axons of human cortical neurons derived from induced pluripotent stem cells (iPS cells). Our 2.7-Å-resolution reconstruction delineates bound water molecules and reveals that axonal microtubules adopt an expanded GTP-like lattice, despite being GDP bound. Using cryo-electron tomography and power spectrum analysis, we find that, unlike in axons, microtubules in undifferentiated iPS cells are compacted. Therefore, lattice expansion is part of neuronal differentiation. Our work provides molecular insights into neurogenesis and has implications for understanding microtubule stability and effector recruitment in neurons.

Microtubules are noncovalent polymers essential for all eukaryotic life. They serve as cellular scaffolds, tracks for molecular motors or signal integrators for diverse pathways[1,2] and participate in basic cellular processes, including cell division, motility and differentiation. Their building block, the α/β-tubulin heterodimer, is a GTPase[3]. α/β-Tubulin heterodimers assemble head-to-tail longitudinally to form protofilaments, which then interact laterally to form the cylindrical structure of the microtubule. The α/β-tubulin dimers that lengthen microtubules through longitudinal interactions are in their GTP-bound form to generate a stabilized cap at the growing microtubule end. However, as GTP–tubulin incorporates into the microtubule, it hydrolyzes to GDP–tubulin, which forms an unstable microtubule lattice. In vitro structural studies of mammalian microtubules showed that GTP hydrolysis controls microtubule lattice conformation, with the stable GTP lattice existing in an expanded state and the unstable GDP lattice existing in a compacted state[4–6]. The expanded microtubule lattice has a wider spacing between tubulin dimers, with a more open interface between them, while the compacted lattice has a tighter arrangement of tubulin dimers. α-Tubulin houses the nonexchangeable GTP-binding

site, termed the N-site, where GTP remains bound without hydrolysis; β-tubulin contains the exchangeable GTP-binding site, referred to as the E-site[7]. During polymerization, catalytic elements contributed in *trans* by the incoming α-tubulin subunit[6,8] stimulate hydrolysis of the E-site GTP. Cryo-electron microscopy (cryo-EM) of in vitro assembled microtubules revealed that GTP hydrolysis at the E-site induces structural rearrangements in the nucleotide-coordinating loops[4,9], which propagate to the longitudinal interface (the polymerization interface between adjacent tubulin dimers within a protofilament), leading to the global transition of the microtubule lattice from an expanded to a compacted state[4,5,9–11].

While we understand the structural transitions associated with tubulin polymerization in vitro, we have yet to visualize a microtubule at atomic resolution in a cell to gain insight into the conformational state of tubulin. Moreover, we do not know the lattice spacing of microtubules in neurons or any other differentiated cell type that undergoes large cytoskeletal-driven morphological changes. Power spectrum analysis of the spacing between tubulin subunits in microtubules of human osteosarcoma derived U2OS cells showed that most of the

[1]Cell Biology and Biophysics Unit, National Institute of Neurological Disorders and Stroke, Bethesda, MD, USA. [2]Biochemistry and Biophysics Center, National Heart, Lung and Blood Institute, Bethesda, MD, USA. ✉e-mail: Antonina@nih.gov

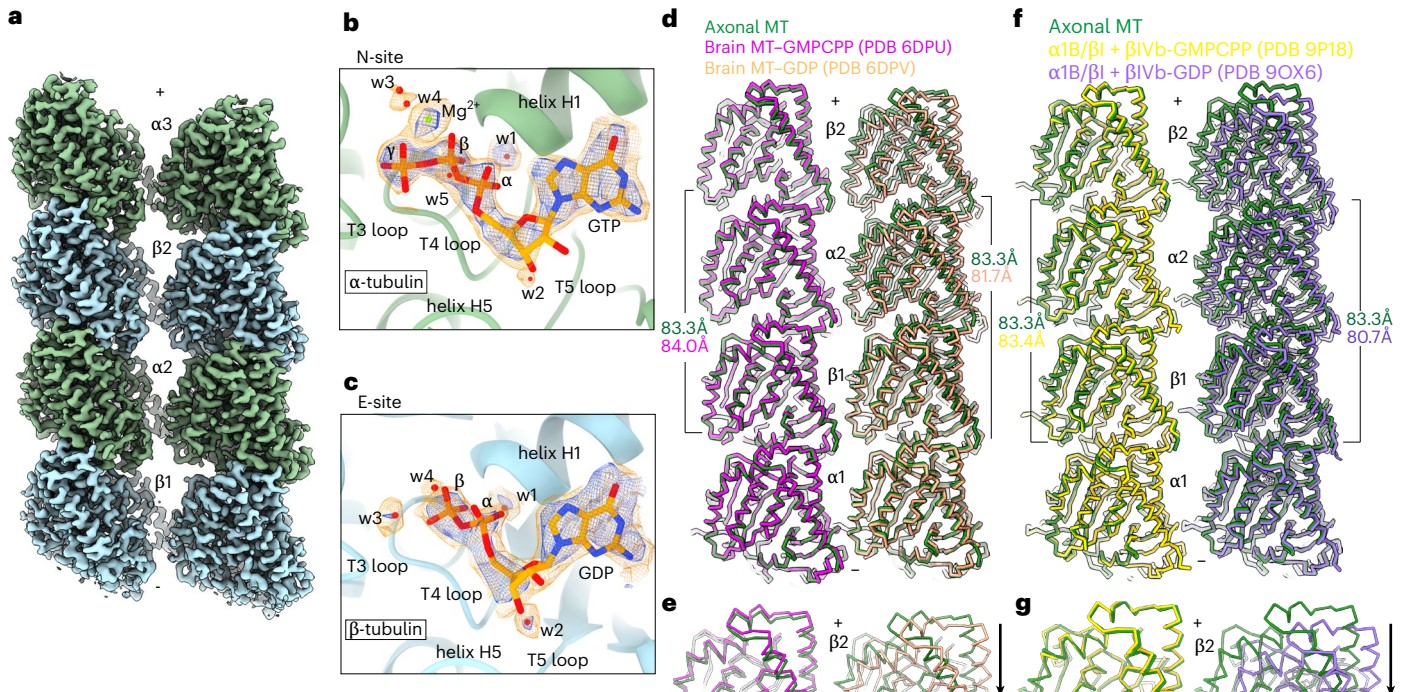

**Fig. 1 | The 2.7-Å-resolution cryo-EM reconstruction of axonal microtubules in human i³Neurons shows an expanded lattice despite a GDP-bound E-site. a**, In situ cryo-EM map of two protofilament from an axonal microtubule in i³Neurons. α/β-Tubulin, green and blue, respectively. '+' and '−' denote microtubule plus and minus ends, respectively (Methods). **b**, N-site, occupied by GTP, in ball-and-stick representation, with a Mg²⁺ ion, shown as a green sphere, and well-ordered waters, shown as red spheres; cryo-EM map for nucleotide, ion and waters at two contour levels: s.d. = 20, blue; s.d. = 12, yellow. **c**, E-site, occupied by GDP, in ball-and-stick representation, with waters as red spheres; cryo-EM map for nucleotide and waters at two contour levels: s.d.= 27, blue; s.d. = 15, yellow. **d**, Protofilament from the axonal microtubule reconstruction (green) superimposed on that from an in vitro assembled undecorated brain GMPCPP–microtubule (left; magenta; PDB 6DPU)[10] or from undecorated brain GDP–microtubule

(right; salmon; PDB 6DPV)[10]. Alignment on the α1-tubulin protomer. Axial dimer rise distances obtained from $C_1$ cryo-EM reconstructions[10] (Methods): 83.3 Å, 84.0 Å and 81.7 Å for axonal microtubules, brain GMPCPP–microtubules and brain GDP–microtubules, respectively. **e**, Detail of the β2-subunit showing lattice compaction, indicated with arrow. **f**, Protofilament from the axonal microtubule reconstruction (green) superimposed on that from an in vitro assembled undecorated α1B/βI + βIVb-GMPCPP–microtubule (left; yellow; PDB 9P18)[11] or α1B/βI + βIVb-GDP–microtubule (right; purple; PDB 9OX6)[11], right. Alignment was performed as in **d**. Axial dimer repeat rise: 83.3 Å, 83.4 Å and 80.7 Å for axonal microtubule, α1B/βI + βIVb-GMPCPP–microtubule and α1B/βI + βIVb-GDP–microtubule, respectively. **g**, Detail of the β2-subunit showing lattice compaction, indicated with arrow.

microtubules are compacted, as expected for GDP lattices, with a small subset in the perinuclear region expanded[12], suggesting local regulation of lattice spacing. Microtubule lattice conformation is the foundational layer of a multilayered regulation of microtubule function[13]. Microtubule-associated proteins (MAPs) and motors distinguish between microtubules that are compacted or expanded[6,12,14–18]; conversely, their binding can alter microtubule lattice spacing[6,10,17,19–21]. The kinesin 3 motor KIF1A has reduced binding to the expanded GTP lattice[15]. In contrast, kinesin 1 KIF5B prefers the expanded lattice while also actively expanding it[14,19]. End binding (EB)-family proteins that regulate microtubule growth prefer the compacted lattice and induce lattice compaction, accelerating GTP hydrolysis[5,10,22]. Targeting protein for Xklp2 (TPX2), which helps nucleate microtubules, prevents lattice compaction and prefers the expanded lattice[21,23]. Tau, involved in Alzheimer disease and frontotemporal dementia[24], regulates motor traffic[25,26] and prefers the compacted microtubule lattice while also compacting it[17]. Tubulin modification enzymes vasohibin (VASH1/2) and metallopeptidase MATCAP1, which detyrosinate microtubules, are also sensitive to lattice spacing[18,27]. The microtubule detyrosination status, in turn, regulates kinesin and dynein motors[28–31] that carry diverse cargo, the microtubule severing enzyme katanin[32] that regulates microtubule density and organization and microtubule EB protein complexes that regulate microtubule growth dynamics[33,34]. Thus, the geometry of the microtubule, specifically its lattice spacing and the conformation of tubulin within its lattice, is central to the control of microtubule functions.

Neuronal differentiation requires a dramatic stabilization and reorganization of microtubules into tightly spaced bundles that scaffold the extension of long-lived axonal processes. In addition to providing structural support, these stable axonal microtubules serve as tracks for the large volume of organized cargo transport required for nervous system function[35]. The molecular basis for the stability of microtubules in the axon or any cell type are unknown, as are changes in microtubule conformation during neuronal differentiation. We used cryo-electron tomography (cryo-ET)-based lattice spacing analysis of microtubules in the intact axons of human induced pluripotent stem cell (iPS cell)-derived cortical neurons and undifferentiated iPS cells and combined this with cryo-EM atomic structure determination to shed light on mechanisms of neurogenesis. We find that the microtubule lattice in the axon expands during neuronal differentiation and reveal the atomic-resolution structure of human microtubules that effectors generate and experience in a live axon, with implications for cargo transport regulation. Our work sets the stage for understanding microtubule function in the intact human neuron at the atomic level.

## Results

### Axonal microtubules are expanded but bound to GDP
To elucidate the molecular mechanism of microtubule assembly and stability in neuronal processes, we used single-particle cryo-EM to obtain a high-resolution structure of the microtubule in the axon of iPS cell-derived human glutamatergic neurons[36,37] at 2.7 Å (Fourier

**Table 1 | Data collection parameters for single-particle cryo-EM and model validation statistics**

| Data collection | |
|---|---|
| Microscope | Krios |
| Detector | K3 |
| Data collection software | SerialEM |
| Energy filter width (eV) | 20 |
| Nominal magnification | ×105,000 |
| Voltage (kV) | 300 |
| Electron exposure (e⁻ per Å²) | 56.57 |
| Exposure rate (e⁻ per pixel per s) | 19.97 |
| Defocus range (µm) | −1.2 to −2.5 |
| Physical pixel size (Å) | 0.824 |
| Time per frame (s) | 0.07 |
| Total exposure time (s) | 2.03 |
| Frames per movie | 29 |
| Movies collected | 3,017 |
| Symmetry imposed | $C_1$ |
| No. of particles | 1,887,666 |
| Map resolution(Å) | 2.7 |
| FSC threshold | 0.143 |
| Map resolution range (Å) | 2.4 to 3.0 |
| Map sharpening $B$ factor (Å²) | −72 |
| **Validation** | **Axonal MT (EMD-70956, PDB 9OX7)** |
| Atomic modeling refinement packages | PHENIX, Coot, ISOLDE |
| Initial model used (PDB code) | 8V4K |
| Model resolution (Å), FSC$_{0.143}$ | 2.7 |
| Model resolution (Å), FSC$_{0.5}$ | 3.0 |
| Model composition | |
| Nonhydrogen atoms | 27,004 |
| Protein residues | 3,452 |
| Ligands | 4 GTP, 4 GDP, 4 Mg²⁺ |
| Waters | 32 |
| $B$ factors (Å²) | |
| Protein | 69 |
| Ligand | 51 |
| Bonds (r.m.s.d.[1]) | |
| Bond lengths (Å) | 0.007 |
| Bond angles (°) | 1.384 |
| **Validation** | |
| MolProbity score | 0.95 (100th percentile) |
| Clashscore | 1.88 (99th percentile) |
| Poor rotamers | 10 (0.28%) |
| EMRinger score | 3.23 |
| $Q$ score | 0.61 |
| Ramachandran plot | |
| Favored | 3,386(98.10%) |
| Allowed | 66(1.90%) |
| Disallowed | 0 |
| Ramachandran $Z$ score | |
| Whole | −0.29 |
| Helix | 0.23 |
| Sheet | −0.46 |
| Loop | −0.35 |
| **Model versus data** | |
| Correlation coefficient (volume) | 0.84 |
| Correlation coefficient (mask) | 0.85 |

[1]Root-mean-square deviation

shell correlation (FSC) = 0.143; Fig. 1a and Extended Data Fig. 1a–e) with a local resolution reaching 2.4 Å in the tubulin core and at tubulin polymerization interfaces (Table 1, Extended Data Fig. 1f and Methods). A previous structure of in situ native cellular microtubules was limited to ~8-Å resolution[38], precluding mechanistic insights into microtubule assembly because the α/β-tubulin protomers, which are highly similar, could not be distinguished and, therefore, did not allow assignment of nucleotide state at the interdimer (E-site) or intradimer (N-site) interface and analysis of polymerization interfaces. Our reconstruction, therefore, marks the highest-resolution structure for a microtubule in situ, laying bare the conformation of the native microtubule within a cell, including the position of well-ordered water molecules at functionally important interfaces.

Our advance in resolution was made possible by innovations in cryo-EM sample preparation and data collection strategies. First, we differentiated iPS cells into 'neurospheres' made of i³Neurons (inducible, integrated and isogeneic)[36,37] and replated them at the edge of EM grids for subsequent extension of radial neuritic projections onto the grid (Extended Data Fig. 1a–c and Methods). While the i³Neurons gradually extend both axonal and dendritic processes, on day 10 after differentiation, the distal processes in these neurospheres (>1 mm beyond the neurosphere core) are exclusively axons, as characterized by their long length and the stereotyped uniform polarity of their microtubules with all plus ends pointed away from the cell body ('plus end out') as determined from cryo-ET analysis[39,40] (Extended Data Fig. 2a,b and Methods). The axonal classification of these processes is also corroborated by microtubule growth directionality assessed by EB1 comet tracking and immunofluorescence staining with axonal and dendritic markers (tau and MAP2, respectively)[41]. This radial configuration allows easy identification of axons and, critically, areas with thin ice, enabling single-particle data collection from hundreds of axons for high-resolution structural determination. An analogous experimental configuration was previously used with human cerebral organoids but was limited to cryo-ET ultrastructural characterization, at nanometer resolution[42]. Somatodendritic processes, with the possible exception of distal dendrites, are not amenable to transmission EM analysis in this configuration because of specimen thickness close to the neurosphere core, which is formed by cell bodies and proximal processes in thick vitreous ice, resulting in low-quality images from mixed cellular compartments.

Our 3.7-Å $C_1$ in situ microtubule reconstruction shows a 13-protofilament microtubule with the α/β-tubulin protomers clearly distinguishable, as evidenced by the different appearance of the S9–S10 loop, which is longer in α-tubulin (Extended Data Fig. 3a and Table 1). Further improvement in resolution was achieved by symmetry expansion of the particle dataset, followed by three-dimensional (3D) local refinement with a mask encompassing two adjacent protofilaments (Extended Data Figs. 1d and 3b and Methods). Our final map at 2.7-Å resolution shows clear side-chain definition (Extended Data Fig. 3c,d and Supplementary Data 1), well-resolved nucleotides in both the N-site and the E-site, as well as well-resolved waters and an active-site Mg²⁺ ion (Fig. 1b,c and Table 1). The α-tubulin N-site is occupied by GTP, with clear density for the Mg²⁺ ion coordinated between the β-phosphate and the γ-phosphate, as well as well-ordered water molecules (Fig. 1b and Extended Data Fig. 4a); the β-tubulin E-site is occupied by GDP, with clear density for water molecules (Fig. 1c and Extended Data Fig. 4b). In vitro assembled GDP-bound mammalian microtubules have compacted lattices characterized by a dimer rise that is less than 82 Å (refs. 4,5,9–11). Remarkably, despite having GDP in the E-site, we find that the axonal microtubule lattice is expanded (Fig. 1d–g). This lattice expansion can be easily visualized by aligning a protofilament from the in situ axonal microtubule with those assembled in vitro from purified brain tubulin[10] and affinity-purified α1B/βI + βIVb tubulin[11] that have an expanded (GMPCPP-bound) or compacted (GDP-bound) lattice (Fig. 1d–g). In the axonal GDP-bound microtubule, the dimer

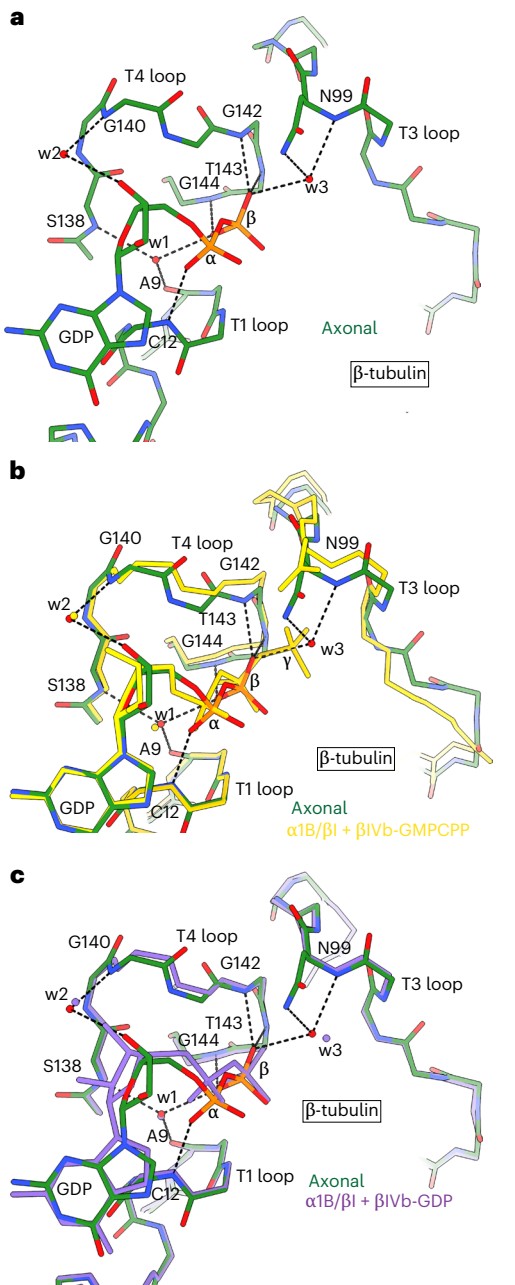

**Fig. 2 | γ-phosphate-sensing elements in the E-site of axonal microtubules are in a GDP-like conformation. a**, Nucleotide coordination by the T1, T3 and T4 loops in axonal microtubules. **b**, Superposition of the axonal (green) and in vitro assembled α1B/βI + βIVb-GMPCPP (yellow; PDB 9P18)[11] microtubule structures showing the displacement of the T3 and T4 loops that sense the γ-phosphate. **c**, Superposition of the axonal and in vitro assembled α1B/βI + βIVb-GDP (purple, PDB 9OX6)[11] microtubule structures shows no significant changes in the position of the T3 and T4 loops. All structures were superimposed on the nucleotide and rigid structural elements that coordinate the nucleotide base in the E-site (residues 10–20 and 221–227). Hydrogen bonds are shown as black dotted lines; water molecules are labeled 'w'.

### Interdimer interface is in GTP-like state

To understand the mechanistic basis for the observed expanded lattice conformation, we used the rigid structural elements that coordinate the nucleotide base (residues 10–20 and 221–227) and the nucleotide itself in the E-site on β-tubulin to align the tubulin dimers from the GDP-bound axonal microtubule structure with in vitro polymerized GMPCPP-bound and GDP-bound microtubule structures. The alignment shows that the T3 loop, which senses the γ-phosphate in GTP, is in a GDP-like conformation in the axonal microtubule (Fig. 2a–c), consistent with the bound GDP in the E-site. In the axonal microtubule GDP-bound structure, the γ-phosphate position is occupied by a water molecule, w3, similar to what is seen in the high-resolution structure of in vitro assembled GDP–microtubules[11] (Figs. 1c and 2a,c and Extended Data Fig. 4a,b), X-ray structures of soluble tubulin bound to GDP or cryo-EM reconstructions of ADP–actin filaments[43–45]. In the absence of the bulkier γ-phosphate, the T3 loop in the axonal microtubule structure is positioned ~1 Å closer to the nucleotide (Fig. 2b,c), as observed in tubulin that is unpolymerized[46] or assembled into microtubules[4,11]. Part of the T4 loop is also in a GDP-like conformation, positioned closer to the nucleotide to make hydrogen bonds with the β-phosphate (Fig. 2b,c). The T1 loop, which coordinates the α-phosphate through multiple hydrogen bonds, has the same conformation in all structures, consistent with its insensitivity to the presence of GDP versus GTP in the E-site (Fig. 2b,c). Interestingly, the T5 loop, which lies at the longitudinal interface between tubulin dimers and is distal to the γ-phosphate, adopts a GMPCPP-like conformation in the axonal microtubule lattice[4,5,9] (Fig. 3a–d). The T5 loop in β-tubulin serves as a nucleotide hydrolysis-sensitive switch that activates tubulin for microtubule assembly[44,46]. It is coupled to the longitudinally adjacent α-tubulin subunit at the plus end through the α-tubulin helix H8 and the T7 loop (Extended Data Fig. 5a). In vitro, upon GTP hydrolysis, the T5 loop moves ~2 Å toward the minus end in concert with the α-tubulin subunit of the adjacent dimer at the plus end, driving lattice compaction[4,9,11] (Fig. 3c,d). In our axonal microtubule structure, the T5 loop, together with T7 and helix H8, is in a GMPCPP-like conformation, displaced ~1.7 Å from the position occupied in a canonical GDP lattice (Fig. 3c,d). In the canonical GDP–microtubule assembled in vitro, V175 in the T5 loop makes tight van der Waals interactions with I332 in helix H10 of the longitudinally adjacent α-tubulin subunit (α2) and S176 H-bonds with residues in strand S9 of the α2-subunit (Fig. 3e and Supplementary Fig. 1). In contrast, in axonal microtubules and GMPCPP–microtubules, V175 and S176 are disengaged from helix H10 and strand S9 (Fig. 3f,g). Similarly, in the canonical GDP–microtubule lattice, T1 and T3 loop residues are engaged in intimate van der Waals and hydrogen-bonding interactions with residues in helix H8 and the T7 loop of the adjacent α-tubulin subunit (α2). These contacts are weakened or broken in the axonal lattice, as well as in the GMPCPP lattice (Extended Data Fig. 5b–d). Thus, tubulin in axonal microtubules exists in a hybrid state with the nucleotide-sensing elements proximal to the γ-phosphate in a GDP-like conformation (Fig. 2), consistent with the GDP occupancy in the E-site, but structural elements distal to the γ-phosphate, which are allosterically linked to the E-site during polymerization in vitro, are in a GTP-like expanded conformation (Fig. 3) that is characteristic of stable microtubule lattices.

GTP hydrolysis in the E-site is catalyzed in *trans* by invariant E254 in helix H8 of the α2-tubulin subunit added during polymerization[8,47]. Upon hydrolysis, in vitro, E254 moves toward the β1-tubulin subunit at the minus end, together with helix H8 and the T7 loop[4–6] (Fig. 4a). This movement is accompanied by the movement of the β-sheet formed by the S7–S10 strands, driving overall lattice compaction. The S7–S10 β-sheet is coupled to helix H8 through invariant F202, which packs against invariant F255 in helix H8 (Fig. 4a). The F202–F255 pair is in a parallel displaced conformation in all existing expanded GMPCPP lattices and a T-shaped or edge-to-face conformation in all compacted lattices. In our axonal microtubule structure, the F202–F255

rise is 83.3 Å, as calculated from the $C_1$ reconstruction (Methods), in contrast to that in GDP-bound microtubules, where it is 81.7 Å and 80.7 Å for brain and α1B/βI + βIVb microtubules, respectively. Instead, the axonal microtubule lattice spacing is more like that of expanded GMPCPP-bound brain and α1B/βI + βIVb microtubule structures, at 84.0 Å and 83.4 Å, respectively[10,11] (Fig. 1d,f).

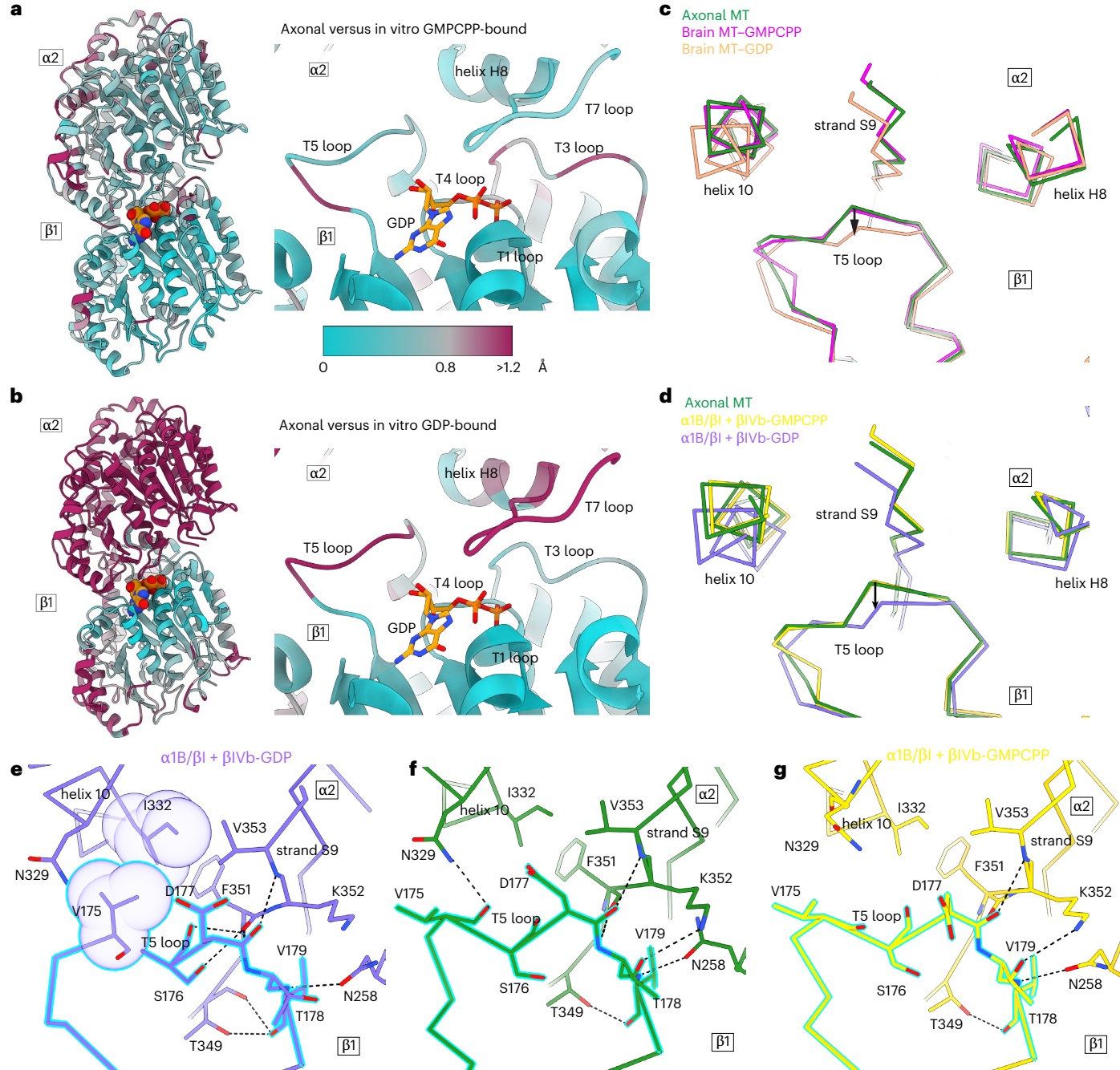

**Fig. 3 | The T5 loop at the longitudinal interface is in a GTP-like conformation in axonal microtubules. a**, Cartoon representation of α2/β1-tubulin protomers at the longitudinal interface, color-coded according to r.m.s.d. between the axonal and the GMPCPP-bound microtubule structure (PDB 9P18)[11]. Structures aligned on the nucleotide and rigid structural elements that coordinate the nucleotide base at the N-site (residues 10–20 and 221–227). **b**, Cartoon representation of α2/β1-tubulin protomers at the longitudinal interface, color-coded according to r.m.s.d. between the axonal and the GDP-bound microtubule structure (PDB 9OX6)[11]. Alignment was performed as in **a. c,d**, The T5 loop at the longitudinal interface is in an extended GTP-like conformation. Models are

shown as Cα-traces. Axonal microtubule, green; brain GMPCPP–microtubule (PDB 6DPU)[10], magenta; brain GDP–microtubule (PDB 6DPV)[10], salmon; α1B/βI + βIVb-GMPCPP–microtubule (PDB 9P18)[11], yellow; α1B/βI + βIVb-GDP–microtubule (PDB 9OX6)[11], purple. Microtubule lattice compaction is indicated with an arrow. **e–g**, Interactions between T5 loop residues (β1) and helix H10 and strand S9 (α2) at the longitudinal interface in the GDP-bound microtubule (PDB 9OX6) (**e**), axonal microtubule (**f**) and GMPCPP-bound microtubule (PDB 9P18) (**g**). Colors are as in **d**. Hydrogen bonds are shown as black dotted lines; van der Waals contacts are shown as spheres. Residues belonging to the β1-protomer T5 loop are outlined in cyan.

phenylalanine pair is in a parallel displaced conformation (Fig. 4b and Extended Data Fig. 6a), consistent with the expanded lattice spacing that we observe. The parallel displaced conformation in the expanded lattice is stabilized by hydrophobic interactions with L259 and L378 (Fig. 4b). Upon GTP hydrolysis, the movement of the catalytic E254 closer to the nucleotide propagates through helix H8 and strand 7,

moving L259 in the same direction (Fig. 4a). In addition to reducing the stabilization of the parallel displaced conformation, this new position of L259 would clash with F255, favoring the transition of F255 to the T-shaped conformation (Fig. 4a). The change in conformation of the F202–F255 phenylalanine pair transmits the compaction at the longitudinal interface to the lateral interface through a relay of invariant

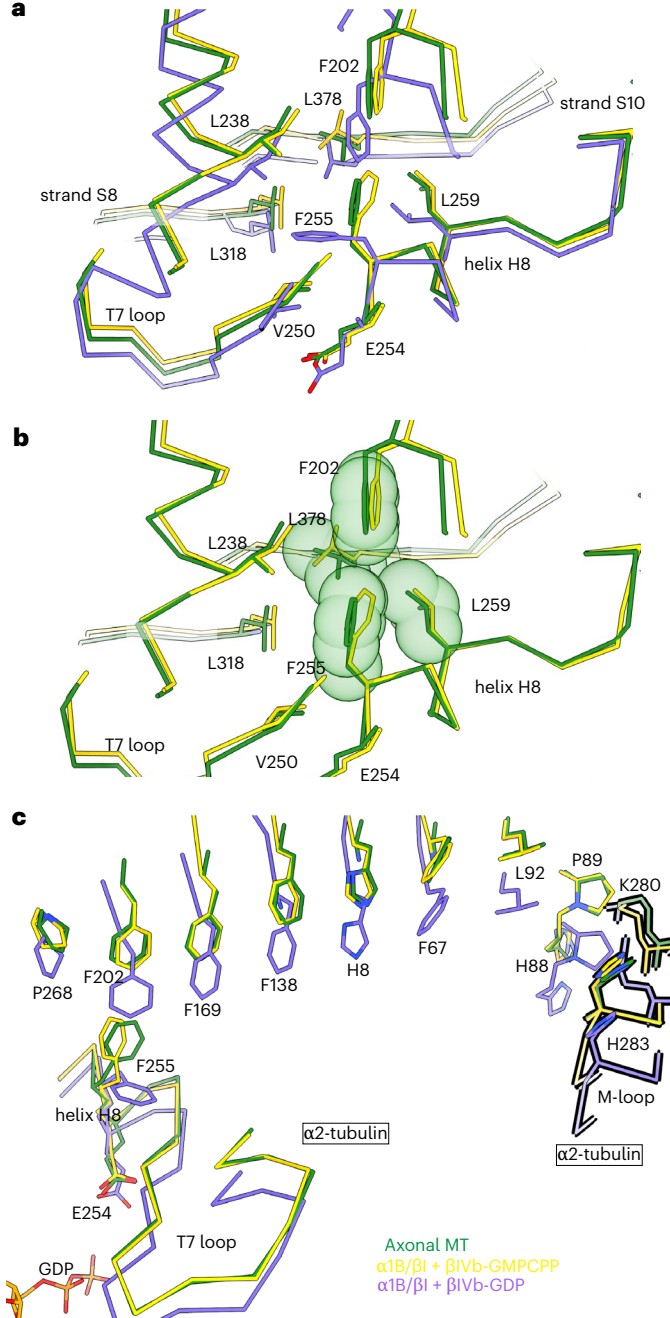

**Fig. 4 | In axonal microtubules, the invariant F202–F255 pair is in a parallel displaced conformation, a signature of expanded lattices. a**, Superposition of axonal, α1B/βI + βIVb-GMPCPP–microtubule and α1B/βI + βIVb-GDP–microtubule structures shows the conformational switch in F255 from parallel displaced to T-shaped conformation, concomitant with lattice compaction. **b**, Contacts that stabilize the F255–F202 pair in α-tubulin in the parallel displaced conformation in the axonal microtubule and GMPCPP–microtubule; van der Waals surfaces are shown as spheres. **c**, Relay of aromatic residues connecting the F202–F255 pair, which is coupled to the catalytic E-site (helix H8 and loop T7) and to the lateral interface. Structures aligned on the nucleotide and rigid structural elements that coordinate the nucleotide base (residues 10–20 and 221–227) at the N-site. Axonal microtubule, green; α1B/βI + βIVb-GMPCPP–microtubule, yellow (PDB 9P18)[11]; α1B/βI + βIVb-GDP–microtubule, purple (PDB 9OX6)[11]. The α-tubulin subunit on the adjacent protofilament is outlined in black and labeled as α2'-tubulin.

aromatic residues F202–F169–F138–H8–F67 crowned by the H2–S3 loop (Fig. 4c, Extended Data Fig. 6b and Supplementary Fig. 1). Residues in the H2–S3 loop (Q85, H88 and E90) interact through several

hydrogen bonds with the microtubule loop (M-loop) of the laterally adjacent tubulin (Fig. 4c and Extended Data Fig. 6b,c), driving the concerted compaction of the lattice. Aromatic π–π stacking interactions are known to form mechanical relays in proteins for transmission of conformation changes[48], including notably for hemoglobin, where flipping of a phenylalanine facilitates its transition between the relaxed and tense states[49]. Aromatic relays have also been used in protein engineering to propagate signals in the nanoscale[50]. Notably, F202 of the phenylalanine pair also connects through a network of hydrophobic interactions to helix H12 (Extended Data Fig. 6d), which is a binding platform for motors and MAPs that could, therefore, allosterically regulate lattice spacing through this network.

## Lattice expansion accompanies neuronal differentiation

To establish whether the expanded microtubule lattice that we observe in the axon is a result of the neuronal differentiation program, which requires exuberant microtubule polymerization and triggers a dramatic increase in the expression of MAPs[51,52], we used cryo-ET coupled with power spectrum analysis to examine lattice spacing in undifferentiated iPS cells and compare it to the microtubule lattice in axons, namely thicker parts of axons not amenable to single-particle data collection (Table 2). The thickness of the iPS cells required focused ion beam (FIB) milling (Fig. 5a and Methods). We analyzed microtubule lattice spacing in Fourier space from their power spectrum[12], with a focus on the layer line close to 40 Å from the equator, which corresponds to the repeat distance between tubulin protomers[53,54] (Extended Data Fig. 7). Specifically, we aligned microtubule segments in 3D to a common reference. The aligned segments were then reextracted into large boxes, microtubule density was masked from the surrounding cellular environment and a two-dimensional (2D) power spectrum for each segment was calculated (Methods, Fig. 5a–e and Extended Data Fig. 7a–d). For each microtubule, the power spectra from nonoverlapping particles belonging to that microtubule were summed to enhance the signal at ~40 Å and a line-plot profile was generated to measure the average lattice spacing (Fig. 5d–f and Extended Data Fig. 7e). To validate our approach, we first tested our procedure on in vitro assembled GDP-bound and GMPCPP-bound brain microtubules for which we collected tomograms using the same parameters as for our cellular analysis (Methods and Table 2). Our analysis showed that, indeed, GDP–microtubules have a compacted lattice with an average spacing between tubulin protomers (either α or β) of 41.6 ± 0.4 Å, while the GMPCPP–microtubules have an expanded lattice with an average spacing of 43.1 ± 0.5 Å (Fig. 5g). Analysis of the lattice spacing of microtubules in the axons of iPS cell-derived human neurons 11 days after differentiation versus that of microtubules in the undifferentiated iPS cells revealed that, in iPS cells, microtubules have more compacted lattices with a mean spacing of 41.9 ± 0.3 Å, close to that of in vitro assembled GDP brain microtubules, while axonal microtubules have an expanded lattice with a mean spacing of 42.6 ± 0.3 Å (Fig. 5f,g), closer to that of in vitro assembled GMPCPP brain microtubules (Fig. 5g). Thus, our power spectrum analysis revealed that the shift of the microtubule lattice in the axon to an expanded state is part of neuronal differentiation.

## Discussion

Motors and MAPs are sensitive to and, in turn, modulate microtubule lattice spacing[13,24]. Our atomic-resolution structure of a microtubule in situ suggests that the collective action of cellular effectors in the axon results in an expanded microtubule lattice that is nonetheless GDP bound. This conformation is associated with stability and different from the compacted lattice expected from in vitro studies. The tubulin dimer within the axonal lattice is in a hybrid state, with structural elements sensitive to the γ-phosphate in the E-site, in a GDP-like conformation, and decoupled from structural elements at the longitudinal interface, which are in a GTP-like conformation. We show that the expanded lattice geometry is acquired during neuronal differentiation. Therefore, our

**Table 2 | Data collection parameters for cryo-ET**

| Data collection | In situ | In situ | In vitro | In vitro |
|---|---|---|---|---|
| | Tomography | Tomography | Tomography | Tomography |
| | Set 1 | Set 2 | Set 1 | Set 2 |
| Microscope | Krios | Krios | Krios | Krios |
| Detector | K3 | K3 | K3 | K3 |
| Data collection software | SerialEM | SerialEM | SerialEM | SerialEM |
| Energy filter width (eV) | 20 | 20 | 20 | 20 |
| Nominal magnification | ×42,000 | ×33,000 | ×42,000 | ×42,000 |
| Voltage (kV) | 300 | 300 | 300 | 300 |
| Electron exposure (e$^-$ per Å$^2$) | From 130 to 160 | From 130 to 150 | 89.4 | 82.6 |
| Exposure rate (e$^-$ per pixel per s) | 17.6 | 14.96 | 21.60 | 18.53 |
| Defocus range (μm) | −3.0 to −4.0 | −3.0 to −5.0 | −4.0 | −4.0 |
| Physical pixel size (Å) | 1.058 | 1.328 | 1.058 | 1.09 |
| Time per tilt angle (s) | From 0.35 to 0.71 | From 0.6 to 1.2 | From 0.24 to 0.48 | 0.24 |
| Frames per tilt angle | From 5 to 10 | From 5 to 10 | From 4 to 8 | 5 |
| Tilt range | 0 to −60; 0 to 60 | 0 to −60; 0 to 60 | 0 to −60; 0 to 60 | 0 to −60; 0 to 60 |
| Tilt increment (°) | 2 | 2 | 2 | 2 |
| Time per frame (s) | 0.07 | 0.12 | 0.06 | 0.07 |

structure reveals the tubulin conformation that motors encounter in the axon while transporting cargo.

In vitro reconstitution studies showed that MAP tau, which is highly abundant in the axon, assembles into cohesive envelopes on brain GDP-bound or GDP–Taxol-bound microtubules (compacting the latter upon binding) but not on expanded GMPCPP–microtubules[17]. The cohesive tau envelopes prevent kinesin 1 access to the microtubule. In contrast, kinesin 1 can access GMPCPP–microtubules even if they are covered with tau[17,26]. It is interesting to speculate that the expanded conformation of the microtubule lattice in the axon might affect tau assembly into cohesive envelopes, with downstream effects on the selective access to motors. However, we note that some MAPs are sensitive not only to lattice spacing but also to the twist between tubulin dimers[10]. Our structure shows that axonal microtubules have a dimer twist similar to that of GDP–microtubules (0.09° versus 0.08° for axonal microtubules and GDP–microtubules, respectively) or Taxol-bound GDP–microtubules[55] (Supplementary Table 1). Therefore, while the axonal and GMPCPP lattice are both expanded, the difference in twist (and protofilament number) makes the in vitro assembled GMPCPP lattice not a suitable analog for axonal microtubules complicating extrapolation of in vitro reconstitution results obtained with GMPCPP–microtubules. When comparing both lattice spacing and twist, axonal microtubules are more like GDP–Taxol-bound microtubules.

Helices H11 and H12 in β-tubulin (Extended Data Fig. 8) form the most common binding platform for microtubule effectors, including kinesin[56,57] and dynein motors[58], tubulin-modifying enzymes[59–61] and MAPs such as tau[62] and TPX2 (ref. 21). The T5 loop, which relays nucleotide-driven conformational changes in the E-site to the longitudinal interdimer interface, is adjacent to helix H11. Our structure reveals that, in axonal microtubules, the T5 loop adopts a conformation characteristic of the expanded lattice, despite the presence of GDP in the E-site. This suggests that, in the axon, effectors allosterically regulate the T5 loop in *trans* to drive changes in lattice spacing. A highly symmetric and specialized mesoscale microtubule structure that is distinguished by its regular decoration with MAPs and microtubule inner proteins (MIPs) is the axonemal doublet. A 4.3-Å ex vivo structure of the axonemal doublet revealed locally expanded microtubule lattices, predominantly in protofilaments in the A-tubule because of the binding of MIPs weaving through the microtubule lattice[20]. Our 3.7-Å

$C_1$ reconstruction does not show any well-resolved nonmicrotubule densities closely associated with the lattice, likely because of the heterogeneous binding of the many MAPs and motors present in the axon. The T5 loop in the expanded ciliary lattice is in a similar conformation to the T5 loop in our axonal microtubule structure (Extended Data Fig. 9). Therefore, the axonemal and axonal microtubule structures illustrate distinct molecular strategies to achieve lattice expansion in these different cellular compartments through the same allosteric element within the tubulin dimer.

Clinically relevant microtubule-targeting agents or drugs also alter microtubule conformation. The drug Taxol (generic name paclitaxel) expands the microtubule lattice when added to mammalian U2OS cells[12]. In vitro, Taxol addition also expands GDP–microtubules assembled from porcine brain tubulin[4,12,55,63], although, in one study, this effect was observed only when the drug was added during tubulin polymerization[55]. The importance of this discrepancy is unclear but could reflect a slow allosteric rearrangement of the microtubule lattice upon Taxol binding. Porcine brain GDP–microtubules bound to Taxol and sparsely decorated with kinesin 1 have an expanded microtubule lattice of 83.3 Å (ref. 64), very similar to the axonal microtubule structure as can be seen from the superposition of the two structures (Extended Data Fig. 10a). GDP–microtubules bound to Taxol and with stoichiometric kinesin 1 decoration also have an expanded lattice of 84.9 Å (ref. 65). Both these Taxol-bound structures have an extended T5 loop (Extended Data Fig. 10b–d) in which, similarly to the GMPCPP-bound and axonal microtubules, V175 and S176 are decoupled from strand S9 and helix H10 of the longitudinally adjacent α2 protomer (Extended Data Fig. 10e and Fig. 3f,g) and the F202–F255 pair is in a parallel displaced conformation (Extended Data Fig. 10f,g). Interestingly, Taxol treatment of hippocampal neurons induces the formation of multiple axon-like processes[66] and promotes axonal regrowth after injury[67]. Our work raises the interesting possibility that these effects are at least partially because of Taxol-induced lattice expansion, promoting the conformational switching in the microtubule lattice in the axon that we now show accompanies neurogenesis, which might lead to a cascade of recruitment of factors sensitive to lattice spacing. Notably, the anterograde motor kinesin 1, which prefers the expanded microtubule lattice[14], selectively accumulates in the emerging axon, coincident with axon specification[68]. We note that while our work reveals a lattice

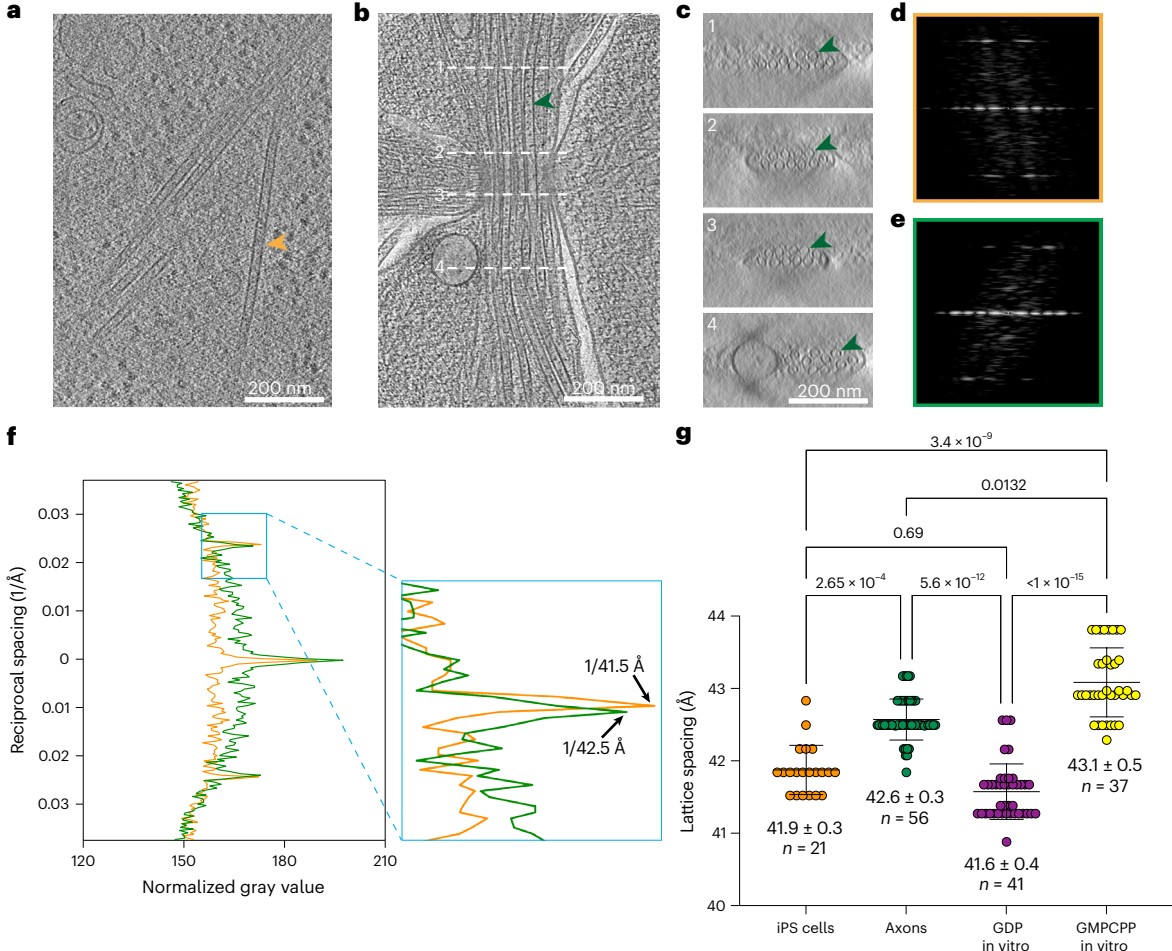

**Fig. 5 | Cryo-ET based in situ lattice analysis shows that microtubules transition from a compacted lattice in iPS cells to an expanded lattice in the axons of i³Neurons. a,b**, Sections of tomograms from iPS cells (**a**) and i³Neuron axons (**b**), with microtubules highlighted by orange and green arrows, respectively. We obtained four tomograms for iPS cells and five tomograms for neurons from four iPS cells and five neurons, respectively. **c**, Axial cross-sections through the tomogram in **b** showing the axonal microtubule bundle. Cross-sections are labeled 1–4. The microtubule highlighted by the green arrow in **b** is also highlighted in the axial cross-sections. Scale bar, 200 nm. **d,e**, Power spectra

of transformed and masked microtubules highlighted by arrows in the iPS cell (**d**) and axon tomograms (**e**). **f**, Overlay of two layer-line plots from the power spectra shown in (**d,e**). Axon, green; iPS cell, orange. Right, zoomed-in view of the line peaks, with peak positions indicated by black arrows. **g**, Microtubule lattice spacing in iPS cells and axons and in vitro assembled brain GDP–microtubules and GMPCPP–microtubules. Data are from four iPS cells and five neurons; *n* indicates number of microtubules analyzed. Bars denote the mean and s.d. The 95% confidence intervals are shown.

conformational switch in the axon, it also remains possible that such a switch is operational in somatodendritic processes, which also undergo microtubule stabilization and reorganization during differentiation. This will be the focus of future studies, together with a detailed analysis of the time dependence of this lattice conformational switch.

Lattice conformation is also sensed by tubulin posttranslational modifications enzymes, notably the VASH12–SVBP enzyme complex and MATCAP1, which preferentially detyrosinate expanded microtubule lattices[18,27]. Recent data also suggest that the tubulin acetylation enzyme αTAT1 prefers the expanded microtubule lattice and MAP7 further enhances acetylation by stabilizing the expanded conformation[69]. Therefore, the microtubule lattice expansion that we observe in the axon during neurogenesis could also lead to encoding of covalent chemical information on the microtubule through the tubulin code, which in turn can further establish feedback loops for microtubule regulation by *trans*-acting factors such as motors and MAPs. The specimen preparation, data collection and processing strategies we present in our study, coupled with the ease of CRISPR-based genetic manipulation in the iPS cell-derived i³Neurons[36,37], as well as the availability as part of the iPS cell Neurodegenerative Disease Initiative (iNDI)[51] of hundreds of i³ lines engineered with variants of genes relevant for

Alzheimer disease, frontotemporal dementia and amyotrophic lateral sclerosis, will enable an atomic-resolution understanding of the effects on microtubule structure of diverse microtubule effectors, in which mutations are a common cause of neurodegeneration and neurodevelopment disorders, and the effects of microtubule-targeting drugs on microtubule structure within the axonal milieu.

## Online content

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

## Methods

### Neurosphere generation

The iPS cell line i11W-mN used in this study was derived from the parental WTC11 (ref. 70) and contains a single copy of doxycycline-inducible mouse NGN2 at the *AAVS1* locus named i11W-mN[36,37]. iPS cells were maintained in wells of six-well cell culture plates (Corning, 3516) freshly coated with Matrigel (Corning, 354277) in StemFlex basal medium (DMEM/F12(Ham) (1:1)) (Thermo Fisher, A33493-01) supplemented with chroman 1 (MedChemExpress, HY-15392) at low density and/or upon passaging for subsequent daily medium changes at 37 °C in an incubator (Thermo Fisher, HERACELL VIOS 160i). Passaging of iPS cells was performed by washing cells once with PBS (Thermo Fisher, 10010-023) followed by 10-min incubation at 37 °C in StemPro Accutase cell dissociation reagent (Thermo Fisher A1110501), which was diluted 1:5 into PBS upon suspension, centrifuged at 300*g* for 5 min (Eppendorf Centrifuge 5810) and resuspended into a small amount of N2 induction medium. Cell density was adjusted to $1 \times 10^5$ cells per ml with N2 induction medium and 60 µl of the cell suspension was added into each well of a 384-well plate (Corning, 4516) that was prewashed with PBS and N2 induction medium. Cells were centrifuged at 500*g* for 10 min to obtain cell clusters and then cultured for 4 days at 37 °C. N2 induction medium was added on days 1 and 2. On days 3 and 4, cells were cultured in 'BP++' medium consisting of BrainPhys (StemCell Technologies, 05790), 1× B27 supplement (Thermo Fisher, A3582801), brain-derived neurotrophic factor (PeproTech, 450-02), NT-3 (PeproTech, 450-03), mouse laminin (Sigma, L2020) and doxycyline (2 µg ml$^{-1}$). On day 4, neurospheres were replated onto the edge of pretreated EM grids.

### Neurosphere differentiation on EM grids

Neurospheres of i³Neurons were differentiated on R2/2 + 2-nm C, Au 200 (Quantifoil) cryo-EM grids (EM Sciences, Q2100AR2-2NM). Specifically, before neurosphere seeding, the grids were plasma-cleaned for 30 s with 15 mA of current at 0.4 mbar of pressure in a PELCO easiGlow glow discharge system. This was followed by sterilization with 70% ethanol for 15 min and washing with double-distilled water (ddH$_2$O) three times. The grids were coated for at least 2 h at 37 °C with 0.1 mg ml$^{-1}$ poly(L-ornithine) (PLO) (Sigma P3655) water solution and then washed four times with ddH$_2$O. Finally, the grids were coated with 15 µg ml$^{-1}$ laminin (Sigma, L2020) in PBS solution for 1 h, followed by two washes with PBS. We used PDMS stencils (Alveole, EM002) to immobilize the grids in a 35-mm glass-bottom Petri dish (MatTek LifeSciences, P35G-1.5-14-C), coated with PLO and laminin. Next, 1.5 ml of BP++ medium was added in the Petri dish containing grids. Most of the BP++ medium in each well of the 384-well plate was aspirated out, leaving about 50 µl. Each neurosphere was gently pipetted into a wide-orifice 200-µl tip and dispensed carefully at the edge of the grid in the Petri dish (Extended Data Fig. 1a). Then, the Petri dish was placed into a 37 °C incubator maintained at 5% CO$_2$ and saturated humidity. The neurospheres were cultured for another 7 days (11 differentiation days total). During this period, the BP++ medium was changed every other day. On day 11, the neurites growing off the EM grid were cut off with a blade just before plunge-freezing in ethane.

### Neurosphere freezing

Excess BP++ medium was blotted from the edge of the EM grid with 595 filter paper 55/20 mm (Ted Pella, 47000-100). The grid was placed inside the Vitrobot environmental chamber operated at 37 °C and 100% humidity. While in the chamber, 3 µl of fresh BP++ medium was added on top of the grid. The consistent volume of liquid on EM grids enabled us to use the same Vitrobot blotting settings for all samples and obtain grids with thin ice. After blotting, each grid was plunge-frozen in liquid ethane. Vitrobot settings were as follows: blot times of 6 s (blot force = 4) and 8 s (blot force = 2) and drain time of 2 s.

### Single-particle cryo-EM data collection

Data were collected on distal parts of the neurosphere, at least 1.75 mm from the neurosphere somatodendritic core center, ensuring that all data were from axonal processes (Extended Data Fig. 1a,b). Data collection was performed on a Krios microscope (Thermo Fisher) equipped with a K3 camera (Gatan) and an energy filter of 20-eV slit width (Table 1). A total of 3,017 videos were acquired at a magnification of ×105,000 in super-resolution mode, with a physical pixel size of 0.824 Å per pixel. The dose rate was 19.97 e$^-$ per pixel per s and the total exposure time was 2.03 s for a 29-frame video with a total accumulated electron dose of 56.57 e$^-$ per Å$^2$. Data were collected using SerialEM (https://bio3d.colorado.edu/SerialEM/)[71] with defocus values ranging from −0.8 µm to −2.5 µm (Table 1). To obtain images with a high signal-to-noise ratio, videos were collected from the thinnest regions of axons.

### Single-particle cryo-EM image processing

Initial image processing was performed in cryoSPARC[72] (Extended Data Fig. 1d) as described previously[73]. Frames were aligned and summed; then, contrast transfer function (CTF) parameters were estimated for each image. Images with a resolution better than 7 Å were used for further processing. Microtubules were automatically traced using Filament Tracer, with microtubule 2D averages as templates. Microtubule segments were extracted into ~600-Å boxes and resampled to a pixel size of 2.472 Å per pixel. The step size between adjacent particle boxes was set to ~80 Å, corresponding to the length of a tubulin dimer. Particles were classified in 2D twice and the best classes were selected for further processing. Next, 3D reconstruction was performed using helical refinement, specifying 13-start pseudohelical symmetry with a rise of 82.5 Å and a twist of 0°, followed by local refinement. Particle coordinates were converted from cryoSPARC to RELION format using pyem and then to FREALIGN[74] using scripts written by R. Zhang (https://github.com/rui−zhang/Microtubule). Using previously published protocols[75], the correct seam location was determined for each microtubule. Particle parameters with the correct seam locations were imported into cryoSPARC and refined locally in 3D, followed by local CTF refinement. The particles were locally refined again to obtain a $C_1$ reconstruction. Helical parameters (helical rise and helical twist per subunit, which describe how one tubulin dimer subunit is related to the adjacent tubulin dimer subunit in the neighboring protofilament) were obtained from the $C_1$ reconstruction using the following relion_helix_toolbox command: relion_helix_toolbox --search --i \$MAP.mrc --cyl_inner_diameter 130.0 --cyl_outer_diameter 330 --angpix \$apix --rise_min 9.00 --rise_max 10.00 --twist_min −28.00 --twist_max −27.00 --z_percentage 0.65 (ref. 76) (Extended Data Fig. 1d). The dimer twist, which describes the rotation that relates one tubulin dimer subunit to the the adjacent tubulin dimer subunit above it in the same protofilament, was obtained from the $C_1$ reconstructions using the following relion_helix_toolbox command: relion_helix_toolbox --search --i \$MAP.mrc --cyl_inner_diameter 130.0 --cyl_outer_diameter 330 --angpix \$apix --rise_min 80.00 --rise_max 85.00 --twist_min −1.00 --twist_max 1.00 --z_percentage 0.65 (ref. 76) (Supplementary Table 1). The dimer distances listed in Fig. 1 were measured from $C_1$ reconstructions using the relion_helix_toolbox program as described by R. Zhang (https://github.com/rui−zhang/Microtubule). For brain microtubules, the dimer rise was obtained from a previous study[10]. To obtain the final two protofilament maps, particles were symmetry-expanded using symmetry parameters measured from the $C_1$ reconstruction (9.61-Å rise and −27.67° twist per subunit) and refined locally again using a mask enclosing four laterally adjacent dimers (Fig. 1a). The nominal resolution of the map was 2.7 Å (FSC cutoff = 0.143) (Extended Data Fig. 1e), with a local resolution ranging from 2.4 to 3.0 Å (Extended Data Fig. 1f and Table 1). Local resolution was estimated using local resolution estimates in cryoSPARC. Local refinement of the map with

a mask around a only single dimer did not improve map quality. The map was sharpened in cryoSPARC with a negative $B$ factor of 72 Å$^2$ (Table 1). The structure was analyzed in UCSF Chimera[77] and UCSF ChimeraX[78]. Figures were generated using UCSF ChimeraX[78] and PyMOL within the SBGrid platform[79].

## Atomic model building and refinement

The model of a brain microtubule (PDB 8V4K)[80] was used as the initial template to build the α1A/βIIb-GDP-bound axonal microtubule structure. Initially, the model was rigidly docked into the cryo-EM map using the UCSF Chimera 'fit in map' tool[77], followed by adjustments in Coot[81] and ISOLDE[82]. The model containing four α/β-tubulin dimers on two protofilaments was iteratively refined in real space using PHENIX, with one round of simulated annealing for each cycle[83]. Waters were identified on the basis of a hydrogen-bonding distance of 2.5–3.0 Å and appropriate geometry. The distance between the β-phosphate and water w3 in the E-site was 2.6 Å (for comparison, the covalent P-to-O distance is 1.6 Å) (Fig. 2a). All software packages are maintained by the SBGrid platform[79]. The final model was validated using the PHENIX 'comprehensive validation' tool (Table 1). The $Q$ scores per residue in α/β-tubulin (model chains A and B) are listed in Supplementary Data 1. $Q$ scores were calculated in ChimeraX[78] using tool QScore[84]. The average $Q$ score was 0.61 (Table 1), consistent with the 2.7-Å overall resolution of the map.

## Sample preparation for cryo-FIB milling

The i11W-mN[36,37] iPS cells were cultured in six-well plates coated with Matrigel in StemFlex basal medium at 37 °C until confluency. Cells were detached from the well by washing once with PBS followed by 10-min incubation in StemPro Accutase cell dissociation reagent at 37 °C. The cell suspension was diluted 1:5 into PBS, centrifuged at 300$g$ for 5 min and resuspended into a small amount of StemFlex basal medium. Cell density was adjusted to $4 \times 10^5$ cells per ml using StemFlex basal medium supplemented with chroman 1.

R2/2 + 2-nm C, Au 200 (Quantifoil) cryo-EM grids were plasma-cleaned and sterilized with 70% ethanol as described above. After three washes with ddH$_2$O, grids were placed into a 35-mm glass-bottom Petri dish (In Vitro Scientific, D35-20-1.5-N) and coated with Matrigel for 15 min at 37 °C. Matrigel solution was removed and 0.5 ml of StemFlex basal medium supplemented with chroman 1 (MedChemExpress, HY-15392) was added to the dish. Next, 1 ml of the cell suspension was added into the dish with grids. Cells were cultured at 37 °C overnight. Grid with cells were plunge-frozen in liquid ethane cooled by liquid nitrogen using Vitrobot operated at 37 °C and 100% humidity. Frozen grids were used for cryo-FIB milling.

## Lamellae preparation

Lamellae from the frozen pluripotent stem cell were prepared using Aquilos 2 cryo-FIB milling (Thermo Fisher). Briefly, grids with stem cells were clipped into the cryo-FIB AutoGrid (Thermo Fisher, 1205101) using the C-Clip (Thermo Fisher, 1036171). Assembled AutoGrids were mounted into the Aquilos AutoGrid Shuttle (Thermo Fisher, 1276420) and transferred into the Aquilos 2 cryo-FIB chamber. Grid overview maps were obtained using scanning EM at ×69 magnification and 2-kV voltage. Cells were coated with an organometallic platinum layer. A new overview map was generated. The milling regions were obtained by two steps: (1) automatic coincidence point and milling angle determination and (2) manual modification of the mill position, area and depth. Relief cutting, rough milling, medium milling, semithin milling and thin milling were conducted automatically. Relief cutting and the rough milling were performed using 1 nA of current and medium milling was performed using 0.5 nA of current. Semithin milling and thin milling were performed at 100 pA, 50 pA and 30 pA. Lamellae were polished manually using 30 pA and 10 pA of current to obtain a thickness ranging from 150 to 250 nm.

## In vitro assembly of microtubules for cryo-EM tomography data collection

Porcine brain GMPCPP-bound microtubules were prepared as described previously[11]. Specifically, porcine brain tubulin (T238P, lot 043, Cytoskeleton) was thawed and spun down to remove aggregates by ultracentrifugation at 279,000$g$ for 10 min at 4 °C. Tubulin was diluted to 2 mg ml$^{-1}$ in 1× BRB80 buffer (80 mM PIPES, 1 mM MgCl$_2$ and 1 mM EGTA) and 1 mM GMPCPP (NU-405L, Jena Bioscience), left on ice for 5 min and then transferred to 37 °C for 1 h to assemble microtubules. Microtubules were spun down at 126,000$g$ for 10 min at 30 °C. The supernatant was discarded and the pellet was washed with warm 1× BRB80 buffer. Microtubules were resuspended in BRB80 buffer and placed on ice for 30 min to depolymerize. GMPCPP was added to 1 mM and the reaction was left on ice for 5 min. The reaction was transferred to 37 °C incubator overnight. The microtubule pellet was spun at 126,000$g$ for 10 min at 30 °C, washed and resuspended in warm 1× BRB80. To measure concentration, microtubules were denatured in 6 M guanidine hydrochloride, followed by measuring absorbance at 280 nm. To prepare GDP-bound microtubules, tubulin was precleared by centrifugation from aggregates as described above and then polymerized at 25 μM in 1× BRB80 buffer and 2 mM GTP for 30 min at 37 °C.

GMPCPP–microtubules were diluted to 1.5 μM in 1× BRB80 buffer, while GDP–microtubules were left undiluted at 25 μM. Next, 5 μl of microtubules were applied on glow discharged lacey grids, UC-A on Lacey 300-mesh Au (01824G, Ted Pella) and allowed to absorb for 30 s. Before the application of microtubules, sixfold-diluted 10-nm BSA gold (25486, EM Sciences) was applied to the EM grids and excess liquid was manually blotted off. The microtubule reactions were blotted with Whatman No. 1 filter paper (Leica Microsystems, 16706440) for 4 s using Leica EM GP2 (Leica Microsystems) with the environmental chamber set to 30 °C and 90% humidity and subsequently plunged into liquid ethane.

## Cryo-EM tomography data collection of in vitro assembled microtubules

Cryo-ET data collection on in vitro assembled brain microtubules was performed using a Krios microscope (Thermo Fisher) equipped with a K3 camera (Gatan) and an energy filter with a 20-eV slit width (Table 2). Two sets of tilt series were acquired in super-resolution mode, with pixel sizes of 1.058 Å per pixel and 1.09 Å per pixel. Tilts were acquired using a bidirectional scheme: 0° to −60°, followed by 0° to 60°, with 2° increments at a magnification of ×42,000. The dose rates were 21.60 and 18.53 e$^-$ per pixel per s and the total exposure time per tilt angle ranged from 0.24 s at 0° tilt to 0.48 s at 60° tilt. The total doses per tomogram were 89.4 and 82.6 e$^-$ per Å$^2$. Data were collected using SerialEM (https://bio3d.colorado.edu/SerialEM/)[71], with defocus values of −4.0 μm. Data collection parameters are listed in Table 2.

## Cryo-EM tomography data collection in cells

Cryo-EM data was collected on a Krios microscope (Thermo Fisher) equipped with a K3 camera (Gatan) and an energy filter with a 20-eV slit width (Table 2). A total of 270 tilt series were acquired at magnifications of ×42,000 (66 tilt series on axon regions) or ×33,000 (139 tilt series on axon regions, 65 tilt series on lamellae) in super-resolution mode, with pixel sizes of 1.058 or 1.328 Å per pixel, respectively. Among these, 56 microtubules from five tomograms collected on five axons were chosen. These tomograms were chosen because of high signal-to-noise ratio and because they contained the entire axon width. A total of 21 microtubules from five tomograms of the lamellae from four different iPS cells were analyzed for lattice spacing. Tilts were acquired using a bidirectional scheme: 0° to −60°, followed by 0° to 60°, with 2° increments. The dose rate was 17.6 or 14.96 e$^-$ per pixel per second and the total exposure time per tilt angle ranged from 0.6 s at 0° tilt to 1.2 s at 60° tilt. The total dose per tomogram ranged from 130 to 150 e$^-$ per

Å². Data were collected using SerialEM (https://bio3d.colorado.edu/SerialEM/)[71], with defocus values ranging from −3.0 µm to −5.0 µm. Cryo-EM data were collected on the distal parts of the neurosphere, at least 1.75 mm from the neurosphere somatodendritic core center, ensuring that all data were collected on axonal processes (Extended Data Fig. 1a,b). Data collection parameters are listed in Table 2.

## Tomography image processing and layer-line analysis

The tilt-series videos were aligned and binned to a physical pixel size using IMOD's alignframes[85]. Tomogram reconstruction was performed in AreTomo[86] for cellular data or reconstructed using batchruntomo for in vitro control[85]. The layer-line analysis procedure followed a previously published method[12]. Reconstructed tomograms were prebinned with a factor of 3 and loaded into Dynamo[87,88]. Microtubules were traced using a filament model (crop along axis). Particles from each microtubule were extracted into 190-pixel (for data with the physical pixel size of 2.116 Å per pixel) or 160-pixel (for data with the physical pixel size of 2.656 Å per pixel) boxes (approximately five dimers long) and averaged to create an initial 3D reference. The particles were then aligned to the 3D reference. Each particle was examined using the dGallery graphical user interface. Nonoverlapping, well-aligned particles were selected to cover the entire microtubule and then reextracted into 800–1,000-pixel boxes (approximately 21 or 26 dimers long). The subsequent image processing steps were performed in Fiji[89] using custom-written scripts, which are available from GitHub (https://github.com/RollmecakLab/Layer-Line-Analysis). The scripts were developed with the assistance of ChIRP, a GenAI chatbot available to National Institutes of Health (NIH) staff (https://bioinformatics.ccr.cancer.gov/btep/classes/chirp-chatbot-for-intramural-research-program). Frames that did not contain microtubule signal were removed from each particle. Paticles collected on lamellae tilted more than 7° in ice (out-of-plane tilt) were rotated to 0° out-of-plane tilt using the rotatevol command in IMOD[85]. All frames per particle were summed along the volume z axis. The summed images were then rotated to align the microtubule long axis along the y axis of the images. Using Fiji's polygon selection tool, microtubules were traced along their perimeter and cropped to remove nonmicrotubule signals. The cropped images were Fourier-transformed and the power spectra of particles from the same microtubule were summed to enhance the signal-to-noise ratio. The lattice spacing was calculated using the following formula:

$$\text{Lattice spacing} = \text{pixel size}\left(\frac{\text{Å}}{\text{pix}}\right) \times \text{box size (pix)}$$
$$\div\left(\text{equator location (pix)} - \text{layer line location (pix)}\right)$$

## Microtubule polarity determination

To determine whether the axons contained uniformly oriented microtubules, we followed a procedure described previously[39]. Briefly, each tomogram was visualized in IMOD[85] and microtubules were traced using open object. Model coordinates were converted into Dynamo[87,88] format and microtubule segments with a box size of 60 nm were cropped with the 'just report' option. Subtomograms were extracted using Subtom version 1.16-32f731b (https://github.com/DustinMorado/subTOM) and in-plane angles were randomized to have a uniform orientation with respect to the missing wedge. All subtomograms were averaged to obtain an initial 3D reference. Subtomograms were iteratively aligned to the average. The alignment mask had the following parameters: box size of '60', radius of '17' and shape 'cylinder'. The alignment was conducted with the following settings: psi_angle_step = (4 4 2), psi_angle_shells = (10 6 6), phi_step = (4 2 2) and phi_agle_shells = 4. Subtomograms for each microtubule were averaged and visualized in IMOD. Polarity was determined on the basis of the rotation of tubulin subunits[40].

## Statistics and reproducibility

Cryo-EM sample preparation and data collection were performed on multiple days to ensure reproducibility. No statistical method was used to predetermine sample size. Tomograms with low signal-to-noise ratio were excluded from analyses because of insufficient data quality. Experiments were not randomized. The investigators were not blinded to allocation during experiments and outcome assessment.

## Reporting summary

Further information on research design is available in the Nature Portfolio Reporting Summary linked to this article.

## Data availability

Cryo-EM maps and atomic models were deposited to the EM Data Bank and PDB under accession numbers EMD-70956 and 9OX7, respectively, with accompanying raw half-maps. Tomograms were deposited to the EM Public Image Archive under the following accession codes: tomograms of axons, EMPIAR-13239; tomograms of iPS cells, EMPIAR-13240; tomograms of in vitro assembled GDP-bound porcine brain microtubules, EMPIAR-13241 and EMPIAR-13242; tomograms of in vitro assembled GMPCPP-bound porcine microtubules, EMPIAR-13243 and EMPIAR-13244. Data and materials can be obtained from the corresponding authors upon request. Source data are provided with this paper.

## Code availability

Scripts used for power spectrum analysis are feely available on GitHub (https://github.com/RollmecakLab/Layer-Line-Analysis).

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

## Acknowledgements

This work used the NIH Multi-Institute Cryo-EM Facility (MICEF). We thank U. Baxa, C. Chen and H. Wang (MICEF) for assistance with data collection and Z. Yu (National Institute of Neurological Disorders and Stroke Cryo-EM Core) for instrument access. This work used the computational resources of the NIH high-performance computing Biowulf cluster (http://hpc.nih.gov). We thank J. Jiang (National Heart, Lung and Blood Institute) for help with pixel size calibration, E. McKenna, M.E. Ward and S. Whittaker (National Institute of Neurological Disorders and Stroke) for advice with iPS cell culture, R. Zhang (Washington University) for sharing the microtubule seam-finding procedure and C. V. Santos and A. Carter (Medical Research Council UK) for sharing protocols for microtubule polarity determination. This research was supported in part by the Intramural Research Program of the National Institute of Neurological Disorders and Stroke and the National Heart, Lung and Blood Institute of the NIH. The contributions of the NIH authors were made as part of their official duties as NIH federal employees, are in compliance with agency policy requirements and are considered works of the US government. However, the findings and conclusions presented in this paper are those of the authors and do not necessarily reflect the views of the NIH or the US Department of Health and Human Services. This work was partially supported by an NIH Director's Innovation Challenge award to A.R.M. A.R.M. is supported by the intramural program of the National Institute of Neurological Disorders and Stroke and the National Heart, Lung and Blood Institute.

## Author contributions

Conceptualization, A.R.M. and E.A.Z. Methodology, E.A.Z., S.S., S.L.S. and A.R.M. Investigation, E.A.Z. (single-particle cryo-EM and cryo-ET) and S.S. (cryo-ET). Data analysis, E.A.Z. (reconstruction and power spectrum analysis) and S.S. (power spectrum analysis). Data interpretation, E.A.Z. and A.R.M. (model and power spectrum analysis) and S.S. (power spectrum analysis). Data acquisition, E.A.Z. (single-particle cryo-EM and in vitro tomography) and S.S. (single-particle cryo-EM and cellular tomography). Visualization, E.A.Z. and S.S. Funding acquisition, A.R.M. Project administration, A.R.M. Supervision, A.R.M. Writing—original draft, A.R.M. and E.A.Z. Writing—review and editing, A.R.M., E.A.Z., S.S. and S.L.S.

## Competing interests

The authors declare no competing interests.

## Additional information

**Extended data** is available for this paper at https://doi.org/10.1038/s41594-026-01787-7.

**Correspondence and requests for materials** should be addressed to Antonina Roll-Mecak.

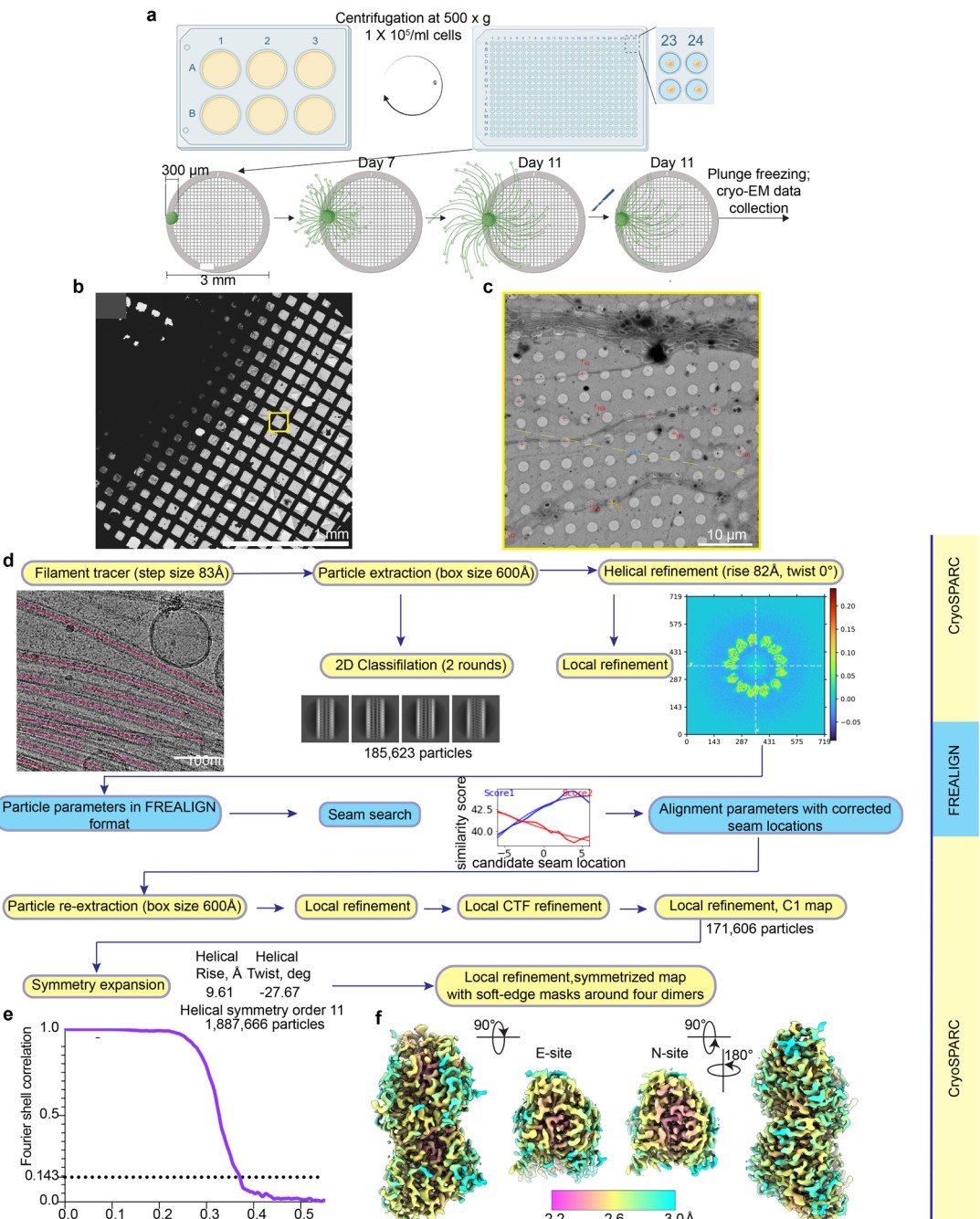

**Extended Data Fig. 1 | Growth of i3Neurospheres on EM grids and cryo-EM image processing. a**, Pluripotent stem cells (iPSC) were grown in 6-well cell culture plates. $1 \times 10^5$/ml cells were transferred to a well of a 384-well plate to form single aggregates (Methods). Aggregates were seeded at the edge of EM grids (one per grid) and induced to differentiate into neurospheres. On day 11 of differentiation, axons extending beyond the EM grid were severed with a razor blade, and grids plunged into liquid ethane for cryo-EM data collection. The panel was created with BioRender.com. **b**, Image of an EM grid with a neurosphere, partially visible in the upper left corner. Single-particle and tomography cryo-EM data were collected from axons, at least 1.75 mm from the neurosphere somatodendritic core center. Single particle cryo-EM data was collected on 9 neurospheres; cryo-ET data was collected on 3 neurospheres. **c**, Higher magnification of a grid square, highlighted in yellow in (b). Red and yellow crosses represent (x, y) positions of acquired images. **d**, Initial and final image processing steps were performed using CryoSPARC[72]. Seam search was conducted in Frealign[74] following previously described protocols[75]. **e**, Fourier shell correlation (FSC) curve for the masked cryo-EM reconstruction. The nominal resolution of the map at $FSC_{0.143}$ is 2.7 Å denoted by the dotted line. **f**, α/β-tubulin dimer color-coded by local resolution. Resolution in the tubulin core reaches 2.4 Å.

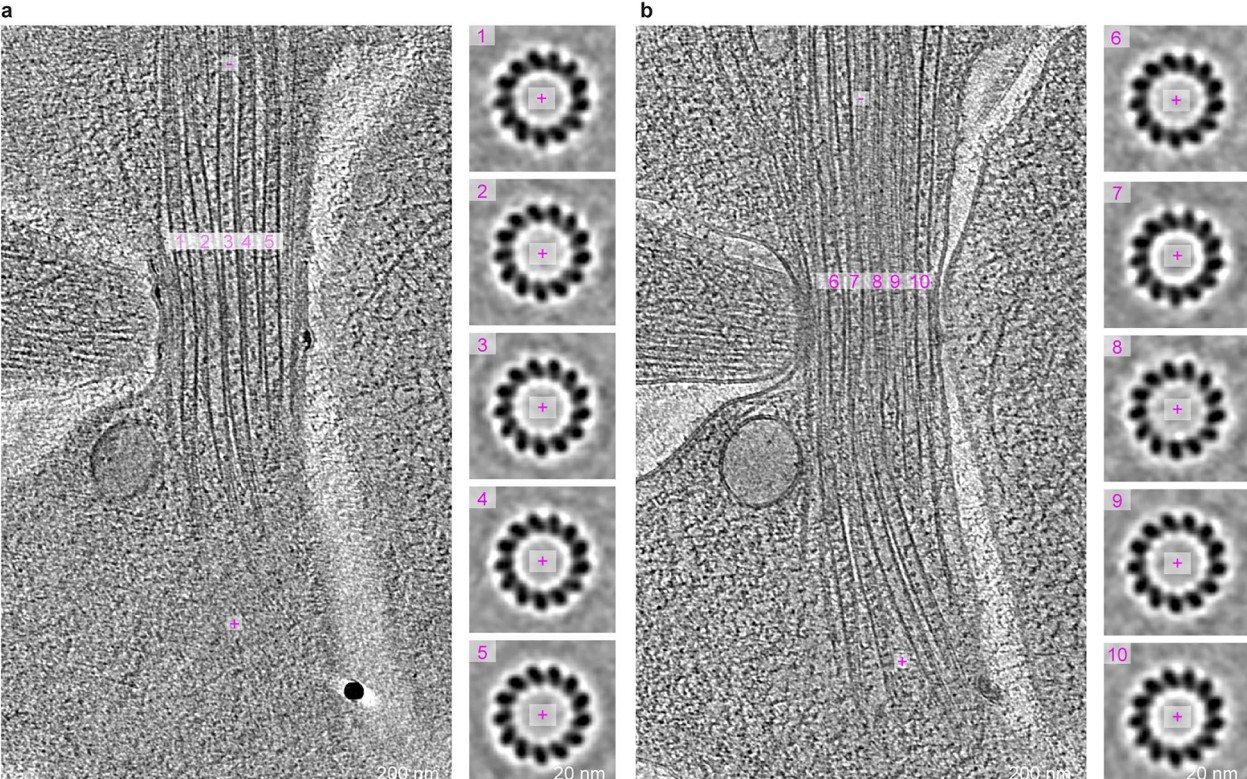

**Extended Data Fig. 2 | Microtubules in axons emanating from i3Neurospheres have uniform polarity with plus-ends out. a**, Tomogram of an axonal microtubule bundle from a day 11 i3Neuron, with microtubules labeled from 1 to 5; plus and minus microtubule ends are indicated with + and – signs, respectively. Small panels on the right show averaged microtubules. All microtubules show the same counterclockwise radial slew of tubulin subunits viewed from the plus end.

Five tomograms from 5 different neurons were collected. **b**, The same tomogram as in (a) showing a second layer of microtubules located ~300 Å away from the first layer shown in (a) within the same bundle. Small panels on the right are averages. All microtubules show the same counterclockwise radial slew of tubulin subunits viewed from the plus end.

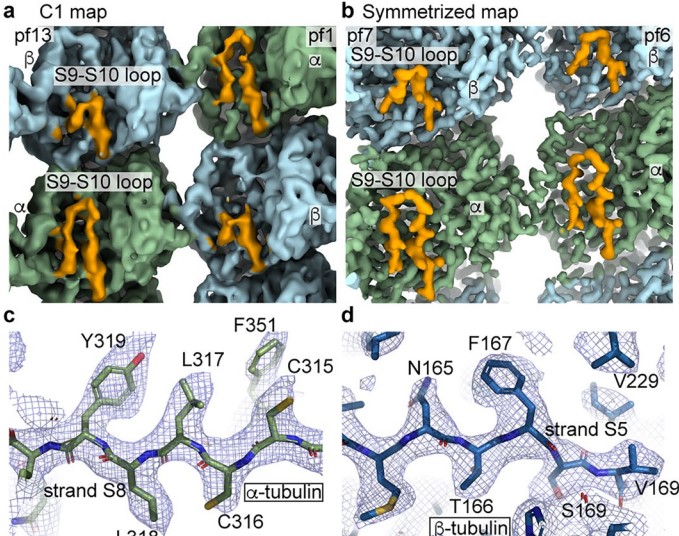

**Extended Data Fig. 3 | Cryo-EM map quality. a**, Seam protofilaments, pf13 and pf1, from the C1 reconstruction. **b**, Protofilaments pf7 and pf6 (opposite of the seam) in symmetrized reconstruction. The S9-S10 loops, orange, is well-resolved and distinguishes α-tubulin (green) which has a longer S9-S10 loop than β-tubulin (blue). **c,d**, β-strands S8 and S5 in α-tubulin and β-tubulin, respectively, with well-resolved sidechains; cryo-EM density, blue mesh.

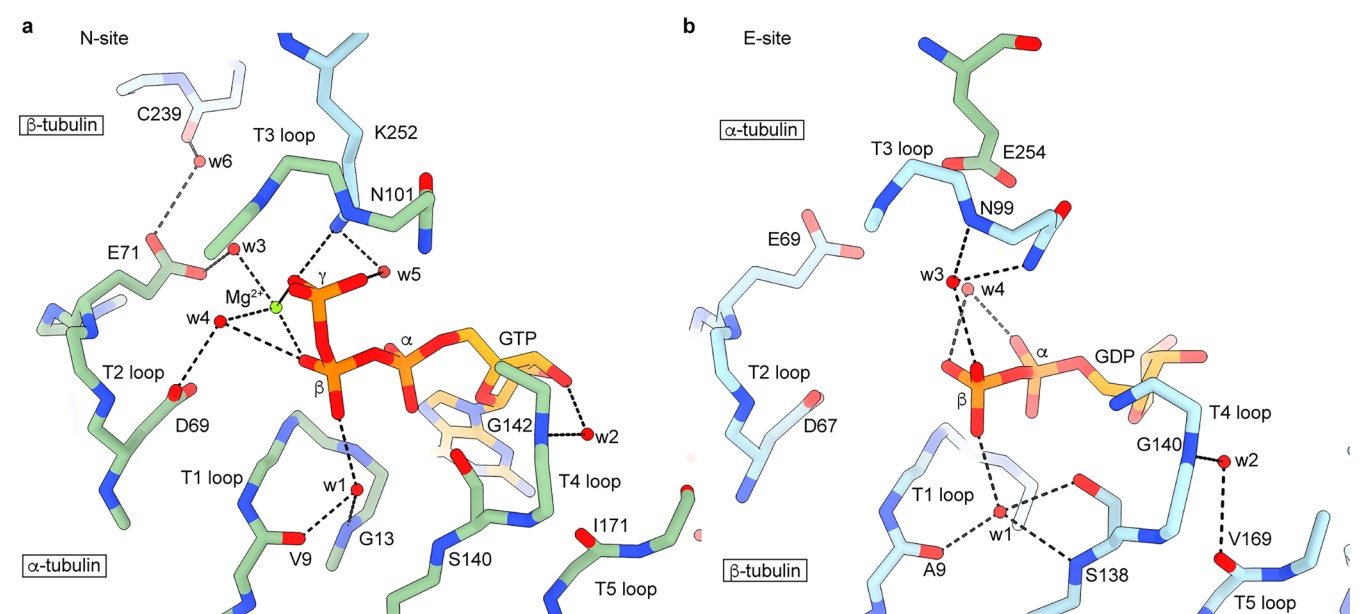

**Extended Data Fig. 4 | Nucleotide coordination in the N- and E-sites. a**, GTP coordination in the N-site. **b**, GDP coordination in the E-site; Mg²⁺ ion, green sphere, water molecules, red spheres. Hydrogen bonds, black dotted lines; α- and β-tubulin, green and blue, respectively.

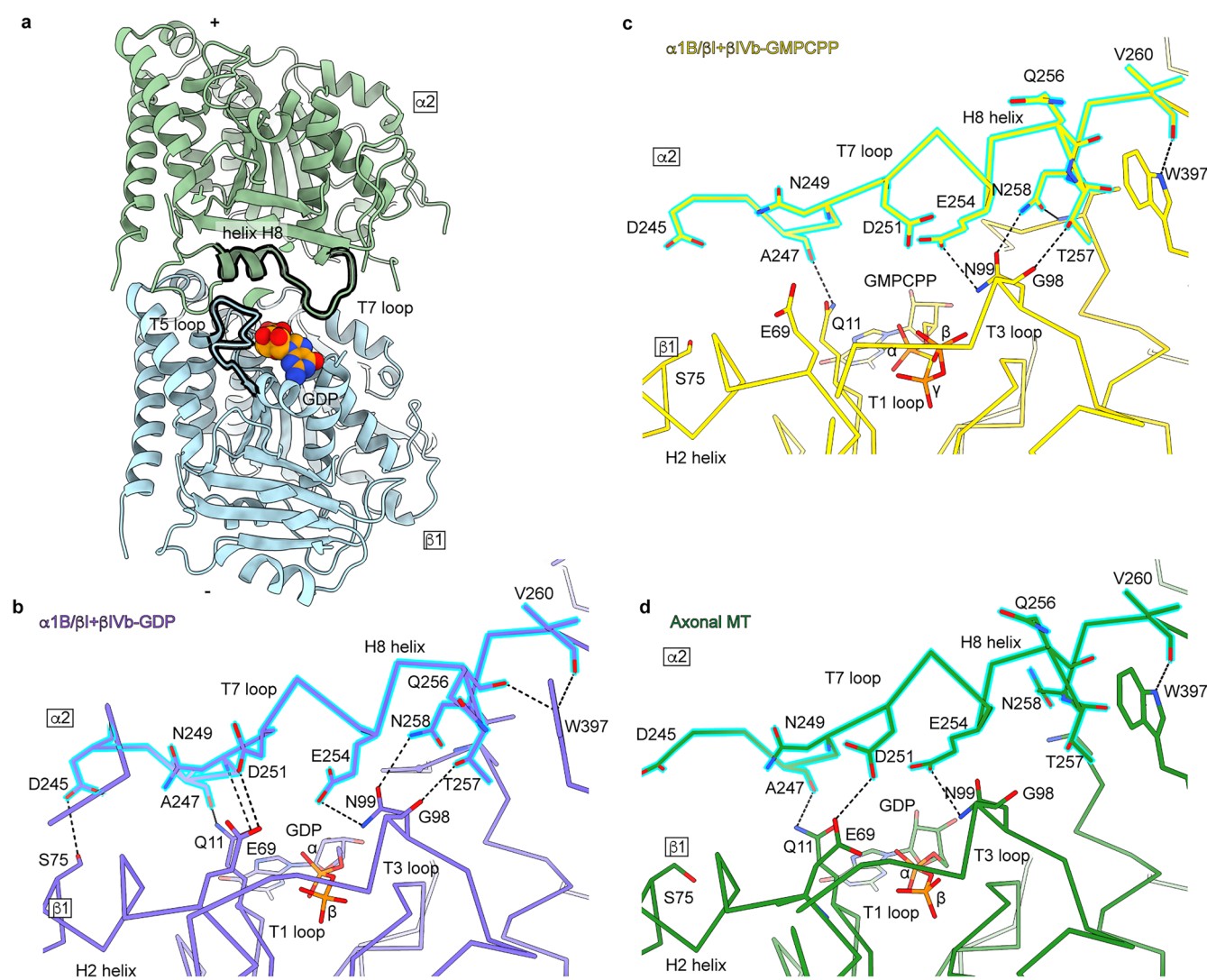

**Extended Data Fig. 5 | Interactions between the T5 loop in β-tubulin and the α-tubulin protomer of the longitudinally adjacent tubulin dimer. a**, The T5 loop in the β1-tubulin protomer is positioned at the longitudinal interface and makes contact with key structural elements (outlined in black) involved in nucleotide-dependent lattice compaction. Microtubule plus and minus ends are indicated with + and – signs, respectively. **b-d**, The T7 loop and helix H8 (α2 protomer) disposition and interactions with the T1 and T3 loops (β1 protomer) at the longitudinal interface of axonal microtubules (**b**) are similar to those observed in GMPCPP-microtubules (**c**) and distinct from GDP-microtubules (**d**) where it makes tighter contacts, both through van der Waals and H-bonds, consistent with the movement of T7 and H8 towards the minus end. T7 and H8 in α2-tubulin protomer are outlined in cyan. H-bonds shown as dotted black lines.

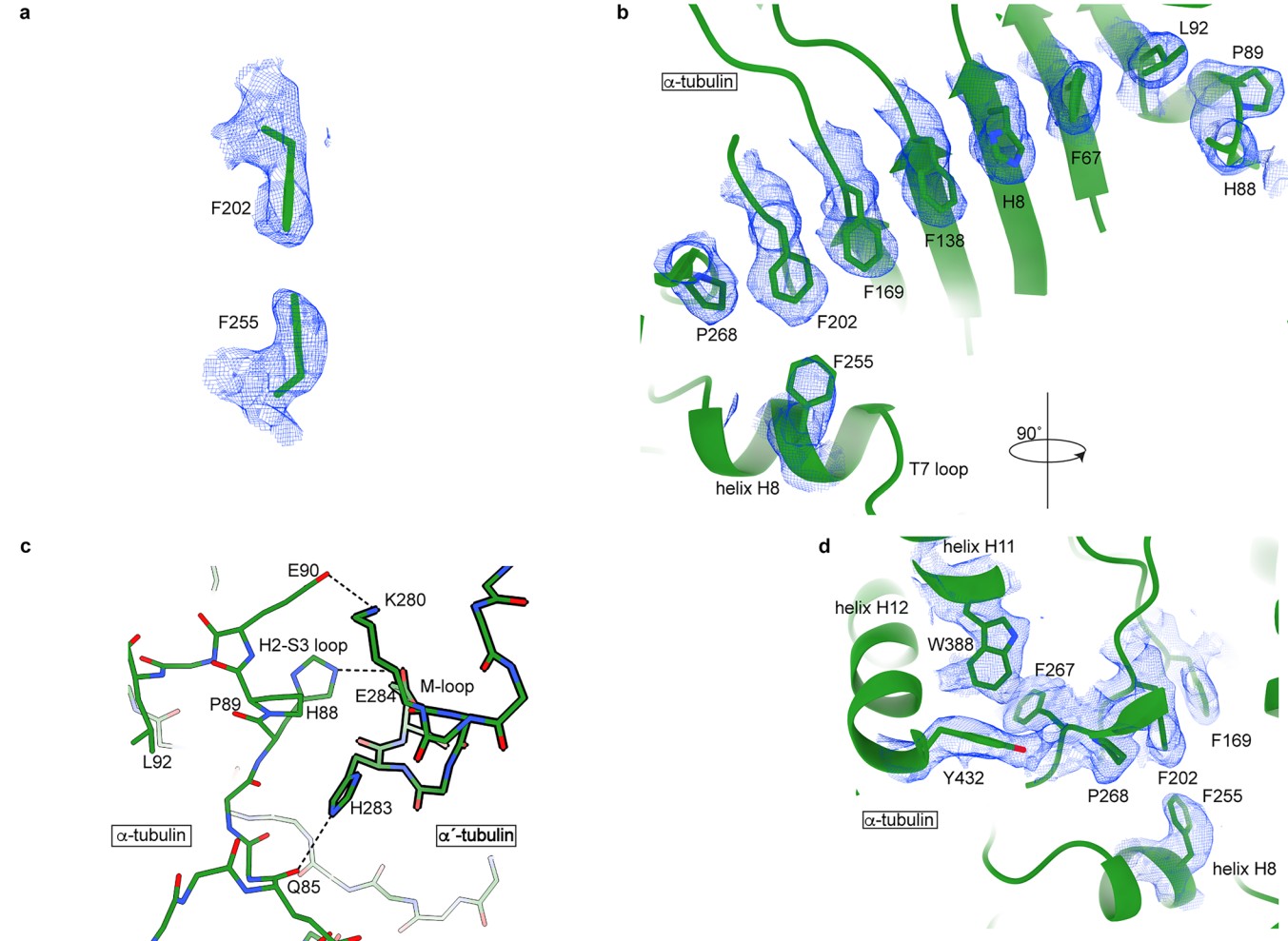

**Extended Data Fig. 6 | The 202–255 phenylalanine pair communicates to the lateral interface through a network of aromatic residues. a**, The F202-F255 pair is in the parallel displaced conformation in the axonal microtubules. **b**, Hydrophobic network linking the E-site to the lateral α-tubulin interface.

Cryo-EM map in blue mesh showing well-defined sidechains. **c**, The α-α'-tubulin lateral interface showing the interaction between the H2-S3 loop and the M-loop in laterally adjacent α-tubulin protomers. The α' protomer is outlined in black. **d**, Hydrophobic network linking F202 to helix H12.

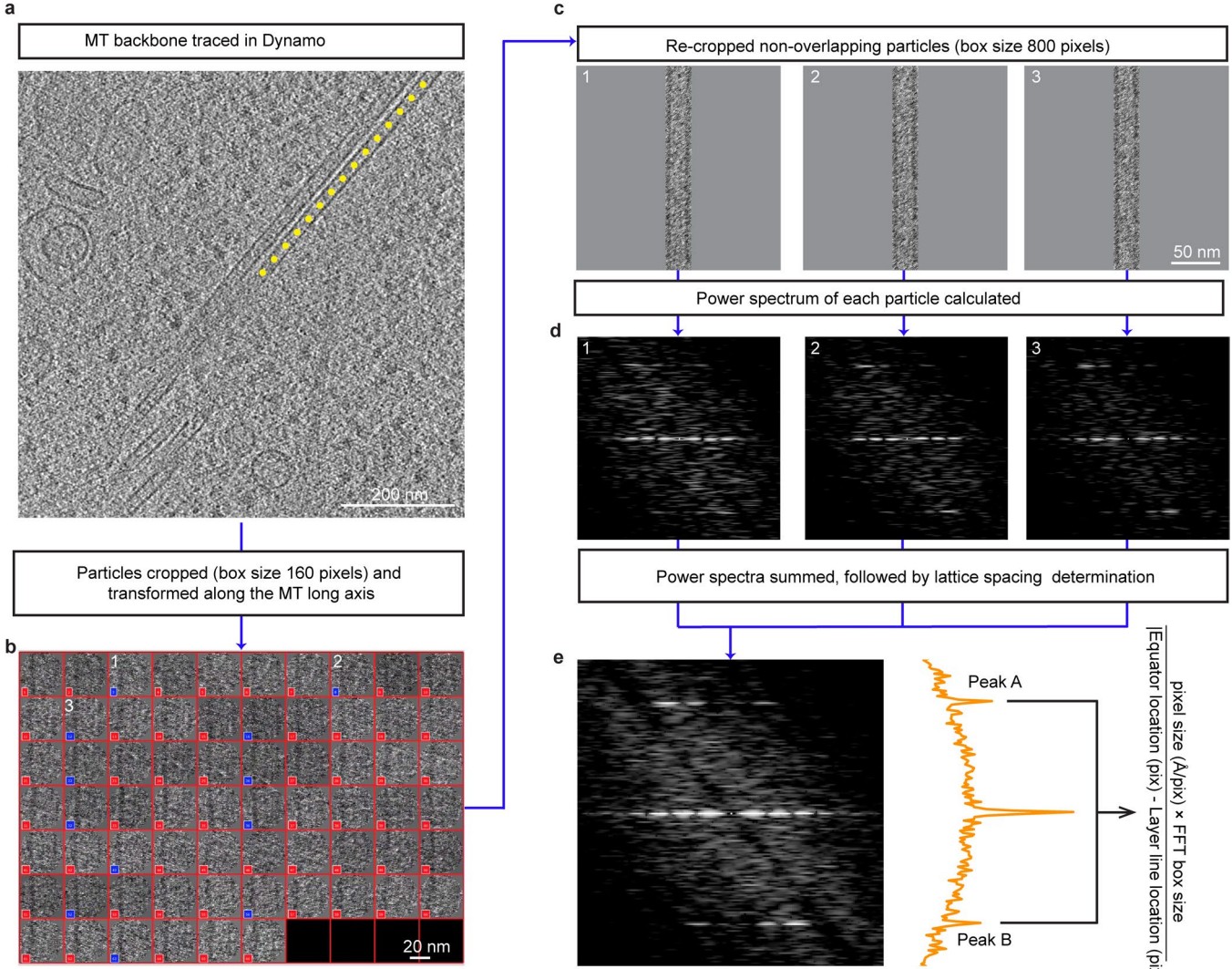

**Extended Data Fig. 7 | *In situ* power spectrum analysis. a**, The microtubule backbone was traced using Dynamo[88] (Methods). **b**, Particles were extracted into small boxes and aligned in 3D to a common reference. Non-overlapping, well-aligned particles selected for further image processing are highlighted by blue boxes in the lower left corner. **c**, Selected particles were re-cropped into 800- to 1000-pixel boxes for power spectrum analysis. Power spectra for particles labeled as 1, 2 and 3 are shown in (**d**). **e**, Power spectra from segments in one microtubule were summed and used to generate a 2D line plot. Microtubule lattice spacing was calculated by measuring the distance between the peaks using the formula shown on the right (Methods).

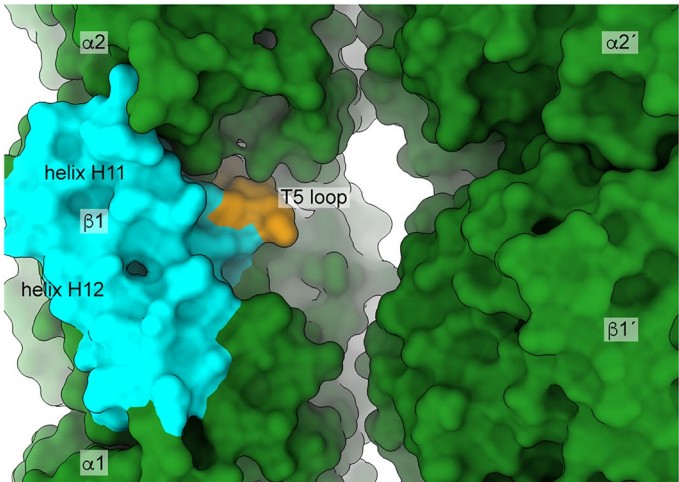

**Extended Data Fig. 8 | The T5 loop in the inter-protofilament groove is solvent-exposed and proximal to the H11 and H12 helices which are binding platforms for many motors and MAPs.** Molecular surface of the axonal microtubule (green) with the T5 loop (yellow) and helices H11 and H12 (cyan). Tubulin subunits in the neighboring protofilaments labeled as α2 and α2′, β1 and β1′.

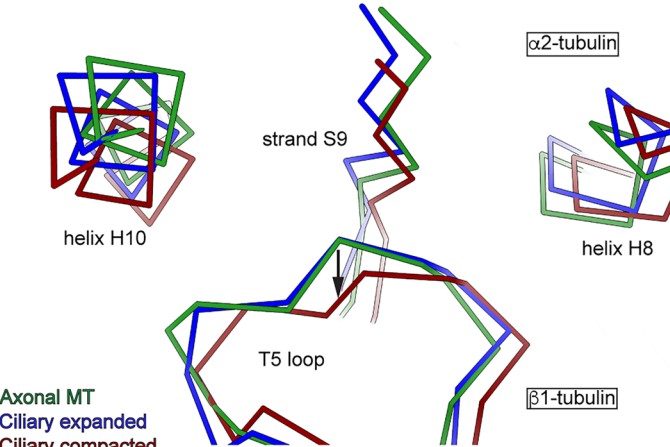

**Extended Data Fig. 9 | The T5 loop in the *in situ* axonal microtubule structure has a conformation similar to that observed in a subset of expanded microtubule protofilaments from axonemes.** The T5 loop in the axonal microtubule structure (green) is in a similar conformation as in the expanded ciliary lattice- (blue; PDB 6U0U ref. 20), ~1.9 Å displaced in comparison with the compacted ciliary lattice (maroon; PDB 6U0U ref. 20). Structures were aligned on the nucleotide and rigid structural elements that coordinate the nucleotide base at the N-site (residues 10–20 and 221–227). Structures shown as Cα-traces.

**a**

Axonal MT vs *in vitro* Taxol-stabilized MT sparsely decorated with kinesin-1(6wvr)

**b**

**c**

Axonal MT
*in vitro* Taxol-stabilized MT sparsely decorated with kinesin-1 (6WVR)

**d**

Axonal MT
*in vitro* Taxol-stabilized MT with stoichiomentric kinesin-1 (9GNQ)

**e**

Taxol-stabilized MT sparsely decorated with kinesin-1(6WVR)

**f** Taxol-stabilized MT sparsely decorated with kinesin-1(6WVR; EMD-21924)

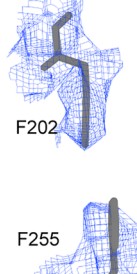

**g** Taxol-stabilized MT with stoichiometric kinesin-1(9GNQ; EMD-51477)

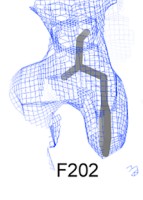
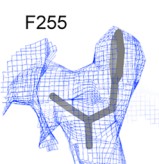

**Extended Data Fig. 10 | The T5 loop in GDP-taxol microtubules sparsely decorated with kinesin-1 and GDP-taxol microtubules bound to kinesin-1 is an expanded conformation and the F202-F255 pair is in a parallel displaced conformation. a**, Cartoon representation of α2- and β1-tubulin protomers at the longitudinal interface, color coded according to root-mean-square-deviation (RMSD) between the axonal and the GDP-bound taxol microtubule structure sparsely decorated with kinesin-1 (PDB 6WVR ref. 64). Structures aligned on the nucleotide and rigid structural elements that coordinate the nucleotide base at the N-site (residues 10–20 and 221–227). **b**, Cartoon representation of α2- and β1-tubulin protomers at the longitudinal interface, color coded according to RMSD between the axonal and the GDP-bound taxol microtubules sparsely decorated with kinesin-1. Structure aligned as in (a). **c**, The T5 loop at the longitdudinal interface in the GDP-bound taxol microtubules sparsely decorated with kinesin-1 is in an extended GTP-like conformation, similar to that in the axonal microtubule. Models shown as Cα-traces; axonal microtubule, green, brain taxol-stabilized microtubule sparsely decorated with kinesin-1, grey.

**d**, The T5 loop at the longitudinal interface in the taxol-stabilized microtubule with stoichiomestric kinesin-1 decoration is in an extended GTP-like conformation, similar to that in the axonal microtubule. Models shown as Cα-traces; axonal microtubule, green, brain taxol-stabilized microtubule with stoichiomestric kinesin-1 decoration, grey. **e**, Interactions between T5 loop residues (β1) and helix H10 and strand S9 (α2) at the longitudinal interface in the taxol-stabilized microtubule sparsely decorated with kinesin-1; H-bonds indicated by black dotted lines. Residues belonging to the β1-protomer T5 loop outlined in cyan **f**, The F202-F255 pair is in the parallel displaced conformation in the expanded lattice of taxol-stabilized microtubules sparsely decorated with kinesin-1. Cryo-EM map (EMD-21924)⁶⁴ in blue mesh showing well-defined side-chains. **g**, The F202-F255 pair is in the parallel displaced conformation in the expanded lattice of taxol-stabilized microtubules with stoichiometric kinesin-1 decoration. Cryo-EM map (EMD-51477)⁶⁵ in blue mesh showing well-defined side-chains.

# Reporting Summary

## Statistics

For all statistical analyses, confirm that the following items are present in the figure legend, table legend, main text, or Methods section.

| n/a | Confirmed | |
|---|---|---|
| ☐ | ☒ | The exact sample size (n) for each experimental group/condition, given as a discrete number and unit of measurement |
| ☐ | ☒ | A statement on whether measurements were taken from distinct samples or whether the same sample was measured repeatedly |
| ☐ | ☒ | The statistical test(s) used AND whether they are one- or two-sided<br>*Only common tests should be described solely by name; describe more complex techniques in the Methods section.* |
| ☒ | ☐ | A description of all covariates tested |
| ☒ | ☐ | A description of any assumptions or corrections, such as tests of normality and adjustment for multiple comparisons |
| ☐ | ☒ | A full description of the statistical parameters including central tendency (e.g. means) or other basic estimates (e.g. regression coefficient) AND variation (e.g. standard deviation) or associated estimates of uncertainty (e.g. confidence intervals) |
| ☒ | ☐ | For null hypothesis testing, the test statistic (e.g. F, t, r) with confidence intervals, effect sizes, degrees of freedom and P value noted<br>*Give P values as exact values whenever suitable.* |
| ☒ | ☐ | For Bayesian analysis, information on the choice of priors and Markov chain Monte Carlo settings |
| ☒ | ☐ | For hierarchical and complex designs, identification of the appropriate level for tests and full reporting of outcomes |
| ☒ | ☐ | Estimates of effect sizes (e.g. Cohen's d, Pearson's r), indicating how they were calculated |

*Our web collection on statistics for biologists contains articles on many of the points above.*

## Software and code

Policy information about availability of computer code

| Data collection | Serial EM v4.1 |
|---|---|
| Data analysis | Cryosparc v4, FREALIGN v9.11, ISOLDE v1.6, Coot v0.9.8.91, PHENIX v1.20.1_4487, ChimeraX-1.6.1, Dynamo v1.1.546, IMOD v5.1.0, EMAN2 v2.99, DeepEMhancer v0.15, Subtom v1.16-32f731b |

For manuscripts utilizing custom algorithms or software that are central to the research but not yet described in published literature, software must be made available to editors and reviewers. We strongly encourage code deposition in a community repository (e.g. GitHub). See the Nature Portfolio guidelines for submitting code & software for further information.

## Data

Policy information about availability of data

All manuscripts must include a data availability statement. This statement should provide the following information, where applicable:

- Accession codes, unique identifiers, or web links for publicly available datasets
- A description of any restrictions on data availability
- For clinical datasets or third party data, please ensure that the statement adheres to our policy

Cryo-EM maps and atomic model are deposited at the EMDB and PDB with accession numbers: EMD-70956 an 9OX7, respectively, with accompanying raw half maps.  Scripts used for power spectrum analysis were deposited on Github (https://github.com/RollmecakLab/Layer-Line-Analysis)

## Research involving human participants, their data, or biological material

Policy information about studies with human participants or human data. See also policy information about sex, gender (identity/presentation), and sexual orientation and race, ethnicity and racism.

| | |
|---|---|
| Reporting on sex and gender | n/a |
| Reporting on race, ethnicity, or other socially relevant groupings | n/a |
| Population characteristics | n/a |
| Recruitment | n/a |
| Ethics oversight | n/a |

Note that full information on the approval of the study protocol must also be provided in the manuscript.

# Field-specific reporting

Please select the one below that is the best fit for your research. If you are not sure, read the appropriate sections before making your selection.

☒ Life sciences  ☐ Behavioural & social sciences  ☐ Ecological, evolutionary & environmental sciences

For a reference copy of the document with all sections, see nature.com/documents/nr-reporting-summary-flat.pdf

# Life sciences study design

All studies must disclose on these points even when the disclosure is negative.

| | |
|---|---|
| Sample size | No sample size calculation was performed. For the layer line analyses conducted in both in vitro and in situ conditions, tens of tomograms were collected, encompassing a broad sampling of microtubules across multiple datasets. The majority of microtubules present within these tomograms were analyzed, providing a sufficiently large and representative dataset to support robust conclusions. |
| Data exclusions | Single-particle cryo-EM images with a resolution less than 7 Å were excluded. Tomograms with low signal to noise ratio or with misaligned frames were excluded. |
| Replication | n/a |
| Randomization | n/a |
| Blinding | n/a |

# Reporting for specific materials, systems and methods

We require information from authors about some types of materials, experimental systems and methods used in many studies. Here, indicate whether each material, system or method listed is relevant to your study. If you are not sure if a list item applies to your research, read the appropriate section before selecting a response.

## Materials & experimental systems

| n/a | Involved in the study |
|---|---|
| ☒ | ☐ Antibodies |
| ☐ | ☒ Eukaryotic cell lines |
| ☒ | ☐ Palaeontology and archaeology |
| ☒ | ☐ Animals and other organisms |
| ☒ | ☐ Clinical data |
| ☒ | ☐ Dual use research of concern |
| ☒ | ☐ Plants |

## Methods

| n/a | Involved in the study |
|---|---|
| ☒ | ☐ ChIP-seq |
| ☒ | ☐ Flow cytometry |
| ☒ | ☐ MRI-based neuroimaging |

# Eukaryotic cell lines

Policy information about cell lines and Sex and Gender in Research

| | |
|---|---|
| Cell line source(s) | Induced pluripotent stem cells (iPSCs) expressing the neuronal transcriptional activator neurogenin 2 (NGN2) under the control of a doxycycline-inducible promoter were obtained from Michael E. Ward lab, and generated as previously described (Wang C et al. Stem Cell Reports. 2017; 9: 1221-1233; Fernandopulle MS et al. Curr Protoc Cell Biol. 2018; 79: e51). Sex-male. |
| Authentication | Cell were authenticated by PCR |
| Mycoplasma contamination | The cell lines tested negative for Mycoplasma. Mycoplasma free cell culture was assessed by the absence of DAPI staining in the cytoplasm of cells. |
| Commonly misidentified lines (See ICLAC register) | These cell lines are not in the database. |

# Plants

| | |
|---|---|
| Seed stocks | n/a |
| Novel plant genotypes | n/a |
| Authentication | n/a |

