## [Peer Review File · Nature Structural & Molecular Biology]

Microtubules in the axon are GDP-bound but adopt a stable GTP-like expanded state

Corresponding Author: Dr Antonina Roll-Mecak

Version 0:

Decision Letter:

7th Jul 2025

Dear Dr. Roll-Mecak,

Thank you again for submitting your manuscript "Microtubules in the axon are GDP-bound but adopt a stable GTP-like expanded state". I apologize for the delay in sharing our decision. I'm writing to let you know that we have decided to send your manuscript for peer review, but several points require your attention before we can proceed with peer review.

I am re-opening the manuscript submission system for you to resubmit your manuscript with all associated files needed for peer review directly, within 2-3 business days if possible. Please follow the link at the bottom of this email to upload the documents listed below. If you have any issues, please reach out to us before completing the submission.

1- We require official wwPDB validation reports for newly described atomic structures, as per journal policy. We also request that authors provide cryo-EM maps, half-maps and models, as well as maps and models obtained from subtomogram averaging if it applies to the work, to help the reviewers in assessing the work. We recommend the use of figshare in our system, which allows for provision of anonymous access links for the referees (<https://www.springernature.com/gp/authors/research-data/figshare-integration>). Alternatively, please upload .zip folders directly with the submission. To ensure the ease of reviewer access to the data, please specify in the Data Availability section where the files can be found (i.e., provide a figshare link or direct the reader to the manuscript files).

2- We want to ensure that the methods and statistics reporting in our papers are of the highest quality. To that end, we ask authors to fill out a Reporting Summary that collects information on experimental design and reagents. If your paper includes ChIP-seq, flow cytometry or MRI data, we ask you take special care to complete those sections of the Reporting Summary as this data will aid greatly in the review of your manuscript. This document can be found by following the link below:

Reporting Summary:

3- In order for us to proceed with peer review, please provide accession numbers and reviewer tokens to access sequencing or proteomics datasets if any unpublished datasets are part of your study. Please add this information to your manuscript file.

4- Lastly, I would like to kindly request that you provide the code used to analyse the data to the reviewers, if newly developed (unpublished) code was used in the work. For the reviewers to evaluate the work adequately, they must be able to test the software/review the code themselves. If you have not yet provided the software, we therefore request that you provide a single compressed zip file containing the software with a readme.txt file or other user manual containing complete instructions for installing and running the software. If appropriate, please provide example data and expected output. Sufficient material should be provided for referees to directly test the performance of the software/algorithm. If the software and materials are small enough to fit in a single compressed zip file under 6MB in size, you may email this file directly to me. If the zip file is between 6 MB and 200 MB, you may upload it to our file transfer site. If necessary, a second zip file up to 200 MB in size can be used to supply the example data. Please let me know if you need to use this option and I'll send you further details. Alternatively, you can upload the code to GitHub and provide us with the link.

Please fill out and return to me the code and software submission checklist that will be made available to editors and reviewers during manuscript assessment. Please note that this form is a dynamic 'smart pdf' and must therefore be downloaded and completed in Adobe Reader, instead of opening it in a web browser.

<https://www.nature.com/documents/nr-software-policy.pdf>

Please use the link below to submit the files. **Please also remember to move forward all other files associated with this version of the paper.**

Link Redacted

Sincerely,
Kat

Katarzyna Ciazynska, PhD
(she/her)
Senior Editor
Nature Structural & Molecular Biology
<https://orcid.org/0000-0002-9899-2428>

Version 1:

Decision Letter:

15th Aug 2025

Dear Dr. Roll-Mecak,

Thank you again for submitting your manuscript "Microtubules in the axon are GDP-bound but adopt a stable GTP-like expanded state". I apologize for the delay in responding, which resulted from the difficulty in obtaining suitable referee reports. Nevertheless, we now have comments (below) from the 3 reviewers who evaluated your paper. In light of those reports, we remain interested in your study and would like to see your response to the comments of the referees, in the form of a revised manuscript.

You will see that while reviewers appreciate the results, they raise several concerns which will need to be addressed in a revision. Specifically, reviewer #2 is asking for a more in-depth structural analysis, based on existing data, and possible further functional follow-ups (points 2 and 3). In line with reviewer #3 comments, we agree that experimentally addressing point 3, about global expansion of MT lattice should be addressed experimentally.

Please be sure to address/respond to all concerns of the referees in full in a point-by-point response and highlight all changes in the revised manuscript text file. If you have comments that are intended for editors only, please include those in a separate cover letter.

We expect to see your revised manuscript within 6 weeks. If you cannot send it within this time, please contact us to discuss an extension; we would still consider your revision, provided that no similar work has been accepted for publication at NSMB or published elsewhere.

Reporting Summary:
<https://www.nature.com/documents/nr-reporting-summary.pdf>

When submitting the revised version of your manuscript, please pay close attention to our <https://www.nature.com/nature-portfolio/editorial-policies/image-integrity> Digital Image Integrity Guidelines. and

to the following points below:

- that unprocessed scans are clearly labelled and match the gels and western blots presented in figures. Please note that all key data shown in the main figures as cropped gels or blots should be presented in uncropped form, with molecular weight markers. While these data can be displayed in a relatively informal style, they must refer back to the relevant figures. These data should be submitted as source data with the last revision, prior to acceptance.
- that control panels for gels and western blots are appropriately described as loading on sample processing controls
- all images in the paper are checked for duplication of panels and for splicing of gel lanes.
- For any revision that includes light microscopy data, we ask our authors to please include a completed light microscopy reporting table [https://www.nature.com/documents/Light_microscopy_reporting_table.xlsx] to ensure the methods are described thoroughly. The table will be available to reviewers and ultimately published should the manuscript be accepted at the journal.

EXTENDED DATA FIGURES

Data availability: this journal strongly supports public availability of data. All data used in accepted papers should be available via a public data repository, or alternatively, as Supplementary Information. If data can only be shared on request, please explain why in your Data Availability Statement, and also in the correspondence with your editor. Please note that for some data types, deposition in a public repository is mandatory - more information on our data deposition policies and available repositories can be found below:

<https://www.nature.com/nature-research/editorial-policies/reporting-standards#availability-of-data>

Link Redacted

Sincerely,

Katarzyna Ciazynska, PhD
(she/her)
Senior Editor
Nature Structural & Molecular Biology
<https://orcid.org/0000-0002-9899-2428>

Referee expertise:

Referee #1: microtubule dynamics

Referee #2: microtubules, cryo-EM

Referee #3: microtubules, neurobiology

Reviewers' Comments:

Reviewer #1 (Remarks to the Author):

In the manuscript titled "Microtubules in the axon are GDP-bound but adopt a stable GTP-like expanded state," the authors present the first high-resolution structure of a microtubule in situ. I will focus my review on the significance of the findings, as the technical aspects of the work are outside my expertise.

In this study, the authors determine an impressively high-resolution structure of microtubules in situ. This is a major advance because, while we have many high-resolution structures of in vitro-polymerized microtubules, data on microtubule structures in their native context are highly limited. This work is a major step forward in obtaining high-resolution structures of microtubules in cells. Although the range of cellular samples in which such structures can currently be determined is limited, I think it is an important step toward determining microtubule structures directly from cells, in particular when combined with the cryoET+power spectrum analysis pipeline.

The high-resolution structure provides mechanistic insight into the expanded lattice state despite GDP being in the active site - a state that is normally associated with a compacted lattice in microtubules that are not bound to associated proteins. In recent years, there has been significant interest in how motors and MAPs can shift the compacted GDP lattice, and in vitro studies on this topic have been conducted (as very nicely summarized in the Introduction). However, whether such an expanded GDP lattice is physiologically relevant has not been clear. The results in this work provide strong evidence for lattice expansion of microtubules in cells and open the possibility of understanding how altering protein expression in different cellular contexts affects microtubule structure.

It would be helpful to see a discussion comparing the structure here with previous in vitro microtubule-GDP-taxol and microtubule-GDP-kinesin structures (albeit those are lower resolution), to highlight common themes and differences.

The cryo-ET+power spectrum analysis to examine lattice spacing in undifferentiated iPSCs and axons correlates well with previous work on stabilizing agents and the upregulation of kinesin-1 during differentiation. However, how well is the distribution of different microtubule populations preserved during sample preparation, and how might this affect the data in Figure 4G?

At the level of significance, I think this work is highly suitable for NSMB for its major technical advance in obtaining a high-resolution microtubule structure in cells, evidence for an expanded GDP-bound lattice in cells, and mechanistic insights into how such an expanded lattice is achieved.

Reviewer #2 (Remarks to the Author):

Dear editors,

Zehr et al. employed single-particle cryo-EM and cryo-ET to examine the microtubule structure in situ. The authors provided structural characterization showing that the axonal GDP-bound microtubules exhibited an expanded GTP-like lattice, whereas microtubules in undifferentiated iPSCs have a compacted lattice. Overall, I support the publication of this work in Nature Structural & Molecular Biology. In particular, the unprecedented 2.7 Å-resolution reconstruction provides molecular details of microtubules within cells. However, I feel that the authors have not fully leveraged the ample information from this near-atomic resolution map, and some gaps need to be addressed before the manuscript is accepted for publication.

Major concerns:

1. With an overall resolution of 2.7 Å and a local resolution of up to 2.4 Å, the authors should be able to identify the

orientation of side chains with good confidence. It is unsatisfying to see that the majority of the structural analyses were performed from a low-resolution perspective (i.e., focusing on backbone movement). It would be more impactful if the authors could conduct a deeper analysis, which might lead to potential functional/cellular assays that would strengthen this manuscript.

One possible direction is to compare their model with those from previous structures of microtubules with GTP-hydrolysis-deficient mutations. Ideally, they would be able to identify the key residue(s) that allow GDP-bound tubulin subunits to adopt the expanded lattice in neuronal microtubules.

2. It is inaccurate to state that paclitaxel expands the GFP-microtubule lattice (Line 274). The drug-induced lattice expansion depends on the microtubule polymerization context and the type of tubulin (Alushin, 2014; Kellogg, 2017; Howes, 2017; Manka, 2018). It will be necessary for the authors to rephrase their proposed hypothesis in the discussion. In addition, how would the paclitaxel treatment affect the inter-subunit spacing of *in situ* microtubules?

3. It is not clear how the GTP hydrolysis would be coupled to the longitudinal interface via the well-ordered network of aromatic residues. The reviewer suggests that the authors clarify their points by rephrasing and/or adding new figures. In addition, based on the authors' near-atomic resolution reconstruction, can they design a mutation that would interfere with the conformational relay through aromatic residues for uncoupling the GTP-hydrolysis from the lattice compaction?

Minor concerns:

1. The author should provide the Q-value, which shows the resolvability of the key residues mentioned in their analysis.

2. In the introduction (Line 92), the authors stated that tubulin PTM enzymes are sensitive to lattice spacing, and the PTM status regulates how other proteins interact with microtubules and microtubule growth dynamics.

This statement is inaccurate, as only VASH1/SVBP (the microtubule detyrosinase) binding is shown to be sensitive to the conformational state of tubulin in the lattice. It is misleading to include the properties not associated with microtubule detyrosination/tyrosination (i.e., microtubule severing enzymes) into this sentence.

Reviewer #3 (Remarks to the Author):

In this manuscript, Zehr and colleagues used the human i3Neurosphere model to investigate the ultrastructure of axonal microtubules using cryo-EM and cryo-FIB milling. These techniques allowed them to resolve the native-state structure of tubulin in neuronal axons at atomic resolution. The authors found that axonal MTs in the human i3Neurosphere exhibit an expanded lattice structure, resembling that of GTP-bound MTs, even though the exchangeable site on β -tubulin was actually bound by GDP. Furthermore, they demonstrated that the inter-dimer interface of GDP-bound tubulin in axonal MTs closely resembles that of GTP-bound tubulin, suggesting that lattice expansion occurs despite GDP occupancy. They find that this expanded lattice conformation is associated with neuronal differentiation.

This manuscript aligns well with the growing interest of applying cryo-ET and cryo-EM in molecular and cellular neurobiology. Importantly, their study offers detailed mechanistic insights at the molecular level into the arrangement of tubulins within the MT lattice in neuronal axons. These findings contribute to a deeper understanding of how MTs support axon formation. While there is no doubt that this work is conceptually significant and is done at a very high technological level, there are several critical points that need to be addressed in the revised manuscript.

Major comments:

1. The approach used to study axonal microtubules in this work closely resembles that of Hoffmann et al. (2021, DOI: 10.7554/eLife.70269). In their study, human brain organoids were used instead of iPSCs to generate axon bundles that grew directly on EM grids, followed by plunge freezing for structural analysis. Although the resolution achieved by Hoffmann et al. was lower, they had already resolved the ultrastructure of axonal MTs in human neurons. This prior work is not cited in the submitted manuscript.

2. In line 111 of the manuscript, the authors claim that their work reveals "the first atomic resolution structure of microtubule that effectors generate and experience *in vivo* in the axon". However, their whole experimental set up is based on cultured iPSCs. So, their work is purely an *in vitro* cell culture study. The authors should also state it clearly.

3. In the section of "Lattice expansion accompanies neuronal differentiation", the authors compared MT lattice in the axon of differentiated neurons to undifferentiated iPSCs and stated that the expansion of MT lattice in neuronal axon is part of the neuronal differentiation program (line 239-240 and figure 4). To my opinion, the evidence is incomplete. Their data certainly indicate that the undifferentiated iPSCs have more compact MT lattice than the axon of differentiated neurons. However, it is possible that this expansion of MT lattice is taking place globally in differentiated i3Neurons. To clarify this, the authors should also analyze MT lattice of the somatodendritic area of the differentiated i3Neurosphere.

4. In the introduction section, the authors stated that Tau, which is an important protein involved in several neurological diseases, preferably binds to compact MTs and is also compacting MT lattice (line 91 & 92). The authors should also briefly discuss how could Tau still bind to MTs, if the axonal MT lattice are in an expanded state which is not preferred by Tau.

5. As already shown by previous literatures that the axon of neurons (as well as dendrite) have both dynamic and stable MTs. The authors should comment whether they could distinguish MTs with different dynamics in their tomography/single

particle data set.

Minor comments:

1. The authors should explain more clearly what is meant by the compact and expanded states of the MT lattice. Clarifying this would help make the manuscript more accessible to a broader readership.
2. Some of the abbreviations is not spelled out. For example, FSC in line 119 is one of the first abbreviation appears at the beginning of the results section, but it is not explained.
3. It happens very often that the authors make a statement but does not reference the figure (e.g. from line 159 to 160, authors should also cite figure 1 at the end of the sentence).
4. There is no scale bar for figure 4a and b.

Version 2:

Decision Letter:

Our ref: NSMB-A51331B

17th Dec 2025

Dear Dr. Roll-Mecak,

Thank you for submitting your revised manuscript "Microtubules in the axon are GDP-bound but adopt a stable GTP-like expanded state" (NSMB-A51331B). It has now been seen by the original referees and their comments are below. The reviewers find that the paper has improved in revision, and therefore we'll be happy in principle to publish it in Nature Structural & Molecular Biology, pending minor revisions to satisfy the referees' final requests and to comply with our editorial and formatting guidelines.

We are now performing detailed checks on your paper and will send you a checklist detailing our editorial and formatting requirements in about 3 weeks. Please do not upload the final materials and make any revisions until you receive this additional information from us.

Sincerely,
Kat

Katarzyna Ciazynska, PhD
(she/her)
Senior Editor
Nature Structural & Molecular Biology
<https://orcid.org/0000-0002-9899-2428>

Reviewer #1 (Remarks to the Author):

The revised manuscript satisfactorily addresses my questions. This is a very nice study, and the findings will have a significant impact on the field.

Reviewer #2 (Remarks to the Author):

The authors have addressed the reviewers' questions/concerns. I support the publication of the manuscript in Nature Structural & Molecular Biology.

Reviewer #3 (Remarks to the Author):

The authors have addressed most of my concerns in the revised manuscript and I appreciate the effort they have made in improving the manuscript. Without asking for more experiments, I think there are still a few points the authors can easily do to make their work more complete. These points are related to their reply to my major comment #3. I understand that only the axonal bundles are vitrified in their work flow, and the neurospheres would indeed be too thick to be vitrified properly via plunge freezing. But this can be achieved alternatively by high-pressure freezing and then combined with the lift-out method, which has been proven possible to prepare lamella from big chunk of materials. I think the authors could very briefly discuss the possibility of doing this in their iPSC-derived neurosphere to resolve MT lattice in the somatodendrite as their future perspective.

In their reply, the authors also mentioned that the somatodendritic area of cultured neurons would be too thick to be imaged via cryo-EM. However, this could be solved by imaging distal dendrites. If they would do this and look into MTs, the outcome would be sufficient to claim if the expansion of MT lattice happens globally in all neurites, or it is a specific thing only for axonal MTs. I am not asking for more experiments, but I suggest to discuss this in the manuscript.

Version 3:

Decision Letter:

4th Mar 2026

Dear Dr. Roll-Mecak,

We are now happy to accept your revised paper "Microtubules in the axon are GDP-bound but adopt a stable GTP-like expanded state" for publication as an Article in Nature Structural & Molecular Biology.

Your paper will be published online soon after we receive proof corrections and will appear in print in the next available issue. You can find out your date of online publication by contacting the production team shortly after sending your proof corrections.

If you have not already done so, we strongly recommend that you upload the step-by-step protocols used in this manuscript to the Protocol Exchange. Protocol Exchange is an open online resource that allows researchers to share their detailed experimental know-how. All uploaded protocols are made freely available, assigned DOIs for ease of citation and fully searchable through nature.com. Protocols can be linked to any publications in which they are used and will be linked to from your article. You can also establish a dedicated page to collect all your lab Protocols. By uploading your Protocols to Protocol Exchange, you are enabling researchers to more readily reproduce or adapt the methodology you use, as well as increasing the visibility of your protocols and papers. Upload your Protocols at www.nature.com/protocolexchange/. Further

information can be found at www.nature.com/protocolexchange/about.

Authors may need to take specific actions to achieve compliance with funder and institutional open access mandates. If your research is supported by a funder that requires immediate open access (e.g. according to [Plan S principles](https://www.springernature.com/gp/open-science/plan-s-compliance) or the [NIH public access policy](https://www.springernature.com/gp/open-science/us-federal-agency-compliance)) then you should select the gold OA route, and we will direct you to the compliant route where possible. Because authors warrant under our subscription licensing terms that they haven't committed to licensing any version of their article under a licence inconsistent with the terms of our agreement – including the applicable embargo period – publication under the subscription model isn't suitable for authors whose funders require no embargo.

Sincerely,

Katarzyna Ciazynska, PhD
(she/her)
Senior Editor
Nature Structural & Molecular Biology
<https://orcid.org/0000-0002-9899-2428>

We appreciate the time and effort dedicated to evaluating our manuscript and thank the reviewers for their strong support of our work, their comments and suggestions. We have now incorporated these into our revised manuscript, both through additional analyses and figures, as well as textual changes. We added two main figures (Fig. 2 and 4), modified Fig. 3, added additional panels in Extended Data Fig. 5 and an additional Extended Data Fig. 10, as well as two Supplementary Tables. We provide a detailed point-by-point response below.

Reviewer 1

“In this study, the authors determine an impressively high-resolution structure of microtubules in situ. This is a major advance because, while we have many high-resolution structures of in vitro–polymerized microtubules, data on microtubule structures in their native context are highly limited. This work is a major step forward in obtaining high-resolution structures of microtubules in cells. Although the range of cellular samples in which such structures can currently be determined is limited, I think it is an important step toward determining microtubule structures directly from cells, in particular when combined with the cryoET+power spectrum analysis pipeline.”

The high-resolution structure provides mechanistic insight into the expanded lattice state despite GDP being in the active site - a state that is normally associated with a compacted lattice in microtubules that are not bound to associated proteins. In recent years, there has been significant interest in how motors and MAPs can shift the compacted GDP lattice, and in vitro studies on this topic have been conducted (as very nicely summarized in the Introduction). However, whether such an expanded GDP lattice is physiologically relevant has not been clear. The results in this work provide strong evidence for lattice expansion of microtubules in cells and open the possibility of understanding how altering protein expression in different cellular contexts affects microtubule structure.”

We thank the reviewer for their support. We were indeed gratified to see that it is possible to achieve this resolution in cells with this system of sample preparation and data analysis, allowing us to visualize an expanded microtubule lattice for the first time in a physiological context, and we

agree with the reviewer that together with the ease of genetic engineering in the iPSC derived i3Neurons, our approach promises to be a powerful in understanding cytoskeletal regulation.

“It would be helpful to see a discussion comparing the structure here with previous in vitro microtubule-GDP-taxol and microtubule-GDP-kinesin structures (albeit those are lower resolution), to highlight common themes and differences.”

Thank you for this suggestion. We have now added additional analysis and an additional figure, Extended Data Figure 10, to address this topic. We now state on pages 14-15:

“Clinically relevant microtubule targeting agents/drugs also alter microtubule conformation. The drug taxol (generic name paclitaxel) expands the microtubule lattice when added to mammalian cells¹². *In vitro*, taxol addition also expands GDP-microtubules assembled from porcine brain tubulin^{4,12,62,63}, although in one study this effect was observed only when the drug was added during tubulin polymerization⁶². The significance of this discrepancy is unclear but could reflect a slow allosteric rearrangement of the microtubule lattice upon taxol binding. Porcine brain GDP-microtubules bound to taxol and sparsely decorated with kinesin-1 have an expanded microtubule lattice of 83.3 Å⁶⁴, very similar to the axonal microtubule structure as can be seen from the superposition of the two structures (Extended Data Fig. 10 a). GDP-microtubules bound to taxol and with stoichiometric kinesin-1 decoration also have an expanded lattice of 84.9 Å⁶⁵. The larger lattice spacing in this case could be related to the additive effects of kinesin-1. Both these taxol-bound structures have an extended T5 loop (Extended Data Fig. 10b-d) in which, similarly to the GMPCPP-bound and axonal microtubules, V175

and S176 are decoupled from strand S9 and helix H10 of the longitudinally adjacent $\alpha 2$ protomer (Extended Data Fig. 10e and Figs. 3f, 3g) and the F202-F255 pair is in a parallel displaced conformation (Extended Data Fig. 10f, 10g). Interestingly, taxol treatment of hippocampal neurons induces the formation of multiple axon-like processes⁶⁶ and promotes axonal regrowth after injury⁶⁷. Our work raises the interesting possibility that these effects are at least partially due to taxol-induced lattice expansion, promoting the conformational switching in the microtubule lattice in the axon that we now show accompanies neurogenesis, and that might lead to a cascade of recruitment of factors sensitive to lattice spacing.”

“The cryo-ET+power spectrum analysis to examine lattice spacing in undifferentiated iPSCs and axons correlates well with previous work on stabilizing agents and the upregulation of kinesin-1 during differentiation. However, how well is the distribution of different microtubule populations preserved during sample preparation, and how might this affect the data in Figure 4G?”

As can be seen in Figure 5, panels b and c, the integrity of the axonal microtubule bundle is well preserved, and we do not see significant changes in lattice spacing within the bundle i.e. all microtubules within this bundle in the distal axon have an expanded lattice (Figure 5g). If the reviewer is referring to different posttranslational modifications for microtubules as different microtubule populations, this will be an interesting future technical challenge to do sub-nm correlative light and cryo-EM in the axon to assign posttranslational modifications to microtubules throughout the axon.

“At the level of significance, I think this work is highly suitable for NSMB for its major technical advance in obtaining a high-resolution microtubule structure in cells, evidence for an expanded

GDP-bound lattice in cells, and mechanistic insights into how such an expanded lattice is achieved.”

We thank the reviewer for their support. We look forward to seeing others using this approach to gain insights into microtubule architecture regulation in the neuron.

Reviewer #2

Zehr et al. employed single-particle cryo-EM and cryo-ET to examine the microtubule structure in situ. The authors provided structural characterization showing that the axonal GDP-bound microtubules exhibited an expanded GTP-like lattice, whereas microtubules in undifferentiated iPSCs have a compacted lattice. Overall, I support the publication of this work in Nature Structural & Molecular Biology. In particular, the unprecedented 2.7 Å-resolution reconstruction provides molecular details of microtubules within cells.”

We thank the reviewer for their support.

“However, I feel that the authors have not fully leveraged the ample information from this near-atomic resolution map, and some gaps need to be addressed before the manuscript is accepted for publication.”

Thank you for this suggestion. We have now added more analyses in our revised manuscript that takes full advantage of the unprecedented resolution we have obtained for microtubules *in situ*.

Major concerns:

1. With an overall resolution of 2.7 Å and a local resolution of up to 2.4 Å, the authors should be able to identify the orientation of side chains with good confidence. It is unsatisfying to see that the majority of the structural analyses were performed from a low-resolution perspective (i.e., focusing on backbone movement). It would be more impactful if the authors could conduct a deeper analysis, which might lead to potential functional/cellular assays that would strengthen this

manuscript. One possible direction is to compare their model with those from previous structures of microtubules with GTP-hydrolysis-deficient mutations. Ideally, they would be able to identify the key residue(s) that allow GDP-bound tubulin subunits to adopt the expanded lattice in neuronal microtubules.

We thank the reviewer for their suggestion; it is very much something we tried to do. However, we are limited in our ability to perform detailed comparisons with other states because of the lower resolutions of the available published structures, even for microtubules assembled *in vitro*. Other than the $\alpha 1\text{B}/\beta\text{I}+\beta\text{IVb}$ structures that we used in our analysis, which are at 3.1 Å and 2.9 Å resolution for GMPCPP (PDB 9bp6) and GDP (PDB 8v2j), respectively, and brain microtubules which are at 3.3 Å and 3.1 Å resolution for GDP (PDB 6dpv) and GMPCPP (PDB 6dpu), respectively, there is only one sub-3Å undecorated microtubule structure, and that is of a GDP-bound E254D microtubule (PDB: 8vt7). This mutant is competent for GTP hydrolysis and has a GDP-like lattice. The undecorated reconstructions of GTP hydrolysis-deficient mutants that the reviewer mentions are at significantly lower resolution than our structure (the E254N mutant structure is at 3.8Å resolution; the E254A mutant reconstruction is reported as 3.4Å but has no deposited atomic model, the E254Q mutant structure is at 3.6Å resolution). Please see list below. The decorated microtubule structure, bound to EB3 for example, is also at ~3.4 Å resolution (PDB 3jar). The positions of many of the side chains (as well as the mainchain in some cases), including at key functional interfaces, are not sufficiently well defined in those cryo-EM maps for the detailed analysis we present in this study. Thus, at present, it is not possible to reliably undertake with those structures the kind of comparative detailed analysis we described in our manuscript.

	Reported resolution (Å)
wt GDP (EMD-25156 , PDB-7SJ7)	3.8
wt GDP-Kinesin (EMD-25157 , PDB-7SJ8)	3.6
GTP-E254A 3-start (EMD-25158)	3.4
GTP-E254A-EB3 (EMD-25159 , PDB-7SJ9)	3.8
GTP-E254N (EMD-25160 , PDB-7SJA)	3.8
GTP-E254N-EB3 (EMD-25161)	5.0

However, as this reviewer points out, our high resolution does give us reliable information on the position of side chains, and thus an ability to dissect microtubule conformation regulation. We have now revised the text and figures to reflect this by providing a more in-depth comparison between our axonal microtubule structure and the existing low 3Å or sub-3Å resolution structures of GDP and GMPCPP microtubules. We modified and moved panels from an extended figure into the main text as Figure 2, which now shows a detailed comparison of the elements that sense nucleotide identity in the E-site (axonal versus GDP *versus* GMPCPP structures). Also, Figure 3 has now three additional panels which show a detailed analysis of the different contacts at the longitudinal interface between the axonal, GDP and GMPCPP structures (Figure 3 e-g). Together, these figures illustrate, at the side-chain conformation level, the hybrid nature of the tubulin conformation in the axonal microtubule, wherein interactions at the longitudinal interface (van der Waals interactions and H-bonds) are very similar to those found in the GMPCPP lattice, while elements that detect the presence of the γ -phosphate are in a GDP-like state. We now state in the main text on pages 9-10:

“In our axonal microtubule structure, the T5 loop, together with T7 and helix H8 are in a GMPCPP-like conformation, displaced ~ 1.7 Å from the position they would occupy in a canonical GDP lattice (Fig. 3c, 3d). In the canonical GDP microtubule assembled *in vitro*, V175 in the T5 loop makes tight van der Waals interactions with I332 in helix H10 of the longitudinally adjacent α -tubulin subunit ($\alpha 2$) and S176 H-bonds with residues in strand S9 of the $\alpha 2$ subunit (Fig. 3e, Supplementary Fig. 1). In contrast, in axonal microtubules, as well as in GMPCPP microtubules, V175 and S176 are disengaged from helix H10 and strand S9 (Fig. 3f, 3g). Similarly, in the canonical GDP-microtubule lattice, T1 and T3 loop residues are engaged in intimate van der Waals and H-bonding interactions with residues in helix H8 and the T7 loop of the adjacent α -tubulin subunit ($\alpha 2$). These contacts are weakened or broken in the axonal lattice, as they are also in the GMPCPP-lattice (Extended Data Fig. 5b-d). Thus, tubulin in axonal microtubules exists in a hybrid state with the

nucleotide-sensing elements proximal to the γ -phosphate in a GDP-like conformation (Fig. 2), consistent with the GDP-occupancy in the E-site, but structural elements distal to the γ -phosphate, which are allosterically linked to the E-site during polymerization *in vitro*, are in a GTP-like expanded conformation (Fig. 3) that is characteristic of stable microtubule lattices.”

Our detailed analysis also led us to identify the Phe202-Phe255 pair that switches from a parallel displaced to a T configuration, as a key structural element in the conversion from an expanded to a compacted lattice, respectively. We have now revised Figure 4 to add additional panels that show the contacts that stabilize the Phe pair in the parallel displaced conformation in the expanded lattice (Figure 4a), and how conformational changes in the catalytic residue E254 lead to the conformational switch in this Phe pair (Figure 4b), which is transmitted through an aromatic relay to the lateral interface. This is now described in the main text on pages 10-11:

“The S7-S10 β -sheet is coupled to helix H8 through invariant F202 which packs against invariant F255 in helix H8 (Fig. 4a). The F202-F255 pair is in a parallel displaced conformation in all existing expanded GMPCPP lattices, and a T-shaped or edge-to-face conformation in all compacted lattices. In our axonal microtubule structure, the F202-255 phenylalanine pair is in a parallel displaced conformation (Fig. 4b, Extended Data Fig. 6a), consistent with the expanded lattice spacing that we observe. The parallel displaced conformation in the expanded lattice is stabilized by hydrophobic interactions with L259 and L378 (Fig. 4b). Upon GTP hydrolysis, the movement of the catalytic E254 closer to the nucleotide, propagates through helix H8 and strand 7, moving L259 in the same direction (Fig. 4a). In addition to reducing the stabilization of the parallel displaced conformation, this new position of L259 would clash with F255, favoring the transition of

F255 to the T-shaped conformation (Fig. 4a). The change in conformation of the F202-F255 phenylalanine pair transmits the compaction at the longitudinal interface to the lateral interface through a relay of invariant aromatic residues F202-F169-F138-H8-F67 crowned by the H2-S3 loop (Fig. 4c, Extended Data Fig. 6b, Supplementary Fig. 1). Residues in the H2-S3 loop (Q85, H88, E90) interact through several H-bonds with the microtubule loop (M-loop) of the laterally adjacent tubulin (Fig. 4c and Extended Data Fig. 6b, 6c), driving the concerted compaction of the lattice. Aromatic π - π stacking interactions are known to form mechanical relays in proteins for transmission of conformation changes⁴⁸, including notably for hemoglobin, where flipping of a phenylalanine facilitates its transition between the relaxed and tense states⁴⁹. Aromatic relays have also been employed in protein engineering to propagate signals in the nanoscale⁵⁰. Notably, F202 of the Phe pair also connects through a network of hydrophobic interactions to helix H12 (Extended Data Fig. 6d), which is a binding platform for motors and MAPs that could therefore allosterically regulate lattice spacing through this network. “

Lastly, we also expanded our structural comparison to that of GDP paclitaxel microtubules bound to kinesin, for which 3Å or better reconstructions are available. This can be now found in new Extended Figure 10 and are described on pages 14-15 of the manuscript. We state:

“Porcine brain GDP-microtubules bound to taxol and sparsely decorated with kinesin-1 have an expanded microtubule lattice of 83.3 Å⁶⁴, very similar to the axonal microtubule structure as can be seen from the superposition of the two structures (Extended Data Fig. 10 a). GDP-microtubules bound to taxol and with stoichiometric kinesin-1 decoration also have an expanded lattice of 84.9 Å⁶⁵. The larger lattice spacing in this case could be related to the additive effects of kinesin-

1. Both these taxol-bound structures have an extended T5 loop (Extended Data Fig. 10b-d) in which, similarly to the GMPCPP-bound and axonal microtubules, V175 and S176 are decoupled from strand S9 and helix H10 of the longitudinally adjacent $\alpha 2$ protomer (Extended Data Fig. 10e and Figs. 3f, 3g) and the F202-F255 pair is in a parallel displaced conformation (Extended Data Fig. 10f, 10g). Interestingly, taxol treatment of hippocampal neurons induces the formation of multiple axon-like processes⁶⁶ and promotes axonal regrowth after injury⁶⁷. Our work raises the interesting possibility that these effects are at least partially due to taxol-induced lattice expansion, promoting the conformational switching in the microtubule lattice in the axon that we now show accompanies neurogenesis, and that might lead to a cascade of recruitment of factors sensitive to lattice spacing.”

2. *It is inaccurate to state that paclitaxel expands the GFP-microtubule lattice (Line 274). The drug-induced lattice expansion depends on the microtubule polymerization context and the type of tubulin (Alushin, 2014; Kellogg, 2017; Howes, 2017; Manka, 2018). It will be necessary for the authors to rephrase their proposed hypothesis in the discussion. In addition, how would the paclitaxel treatment affect the inter-subunit spacing of in situ microtubules?*

The reviewer brings up an important point regarding the effect of paclitaxel. *In vitro*, Alushin *et al.* (Alushin *et al.*, Cell, 2015) reported an expanded lattice for the GDP-taxol microtubules that is similar to that of the GMPCPP microtubule structure reported in the same publication. An expanded microtubule lattice was also observed in other *in vitro* studies (Kellogg *et al.*, JMB 2017; de Jager *et al.* JCB 2025; Estevez-Gallego *et al.*, Elife 2020). Kellogg *et al.* (Kellogg *et al.* 2017) reported an expanded lattice only when paclitaxel was added during polymerization, but not if added after microtubule assembly. This discrepancy between the studies suggest to us that paclitaxel acts allosterically, and if the microtubules are already assembled, the lattice

rearrangements are slow and very sensitive to paclitaxel stoichiometry. In cells, de Jager *et al.* (*de Jager et al. JCB 2025*) showed that paclitaxel addition expands the microtubule lattice, as documented by layerline analysis after addition of paclitaxel to U2OS cells. We have now expanded our discussion to cover in more detail the effects of taxol. All our analyses are focused on mammalian microtubules and not *S. cerevisiae* microtubules which have different lattices and display very different microtubule dynamics parameters compared to mammalian microtubules. We have made sure to specify this in our discussion. We now state:

“Clinically relevant microtubule targeting agents/drugs also alter microtubule conformation. The drug taxol (generic name paclitaxel) expands the microtubule lattice when added to mammalian U2OS cells¹². *In vitro*, taxol addition also expands GDP-microtubules assembled from porcine brain tubulin^{4,12,62,63}, although in one study this effect was observed only when the drug was added during tubulin polymerization⁶². The significance of this discrepancy is unclear but could reflect a slow allosteric rearrangement of the microtubule lattice upon taxol binding.”

In our revised manuscript we now also compare our axonal microtubule structure with that of taxol-stabilized microtubules bound to kinesin. These analyses show that the axonal microtubule structure is very similar to the expanded lattice of GDP paclitaxel microtubules bound to kinesin, reflected in the expanded conformation of the T5 loop, as well as the conformation of the loops that sense the nucleotide (Extended Data Fig. 10). We now state in the expanded discussion:

“Porcine brain GDP-microtubules bound to taxol and sparsely decorated with kinesin-1 have an expanded microtubule lattice of 83.3 Å⁶⁴, very similar to the axonal microtubule structure as can be seen from the superposition of the two structures (Extended Data Fig. 10 a). GDP-microtubules bound to taxol and stoichiometric kinesin-1 decoration also have an expanded lattice of 84.9 Å⁶⁵. The larger lattice spacing in this case could be related to the additive effects of kinesin-

1. Both these taxol-bound structures have an extended T5 loop (Extended Data Fig. 10b-d) in which, similarly to the GMPCPP-bound and axonal microtubules, V175 and S176 are decoupled from strand S9 and helix H10 of the longitudinally adjacent $\alpha 2$ protomer (Extended Data Fig. 10e and Figs. 3f, 3g) and the F202-F255 pair is in a parallel displaced conformation (Extended Data Fig. 10f, 10g). Interestingly, taxol treatment of hippocampal neurons induces the formation of multiple axon-like processes⁶⁶ and promotes axonal regrowth after injury⁶⁷. Our work raises the interesting possibility that these effects are at least partially due to taxol-induced lattice expansion, promoting the conformational switching in the microtubule lattice in the axon that we now show accompanies neurogenesis, and that might lead to a cascade of recruitment of factors sensitive to lattice spacing.”

3. It is not clear how the GTP hydrolysis would be coupled to the longitudinal interface via the well-ordered network of aromatic residues. The reviewer suggests that the authors clarify their points by rephrasing and/or adding new figures. In addition, based on the authors' near-atomic resolution reconstruction, can they design a mutation that would interfere with the conformational relay through aromatic residues for uncoupling the GTP-hydrolysis from the lattice compaction?

We added an additional panel that illustrates the role of Phe255 flipping in response to GTP hydrolysis, as well as expanded our description of the aromatic residue relay to the lateral interface. These can be found in revised Fig. 4 and Extended Fig. 6 and described on pages 10-11 of the revised manuscript. We now state:

“Upon GTP hydrolysis, the movement of the catalytic E254 closer to the nucleotide, propagates through helix H8 and strand 7, moving L259 in the same direction (Fig. 4a). In addition to reducing the stabilization of the parallel displaced conformation, this new

position of L259 would clash with F255, favoring the transition of F255 to the T-shaped conformation (Fig. 4a). The change in conformation of the F202-F255 phenylalanine pair transmits the compaction at the longitudinal interface to the lateral interface through a relay of invariant aromatic residues F202-F169-F138-H8-F67 crowned by the H2-S3 loop (Fig. 4c, Extended Data Fig. 6b, Supplementary Fig. 1). Residues in the H2-S3 loop (Q85, H88, E90) interact through several H-bonds with the microtubule loop (M-loop) of the laterally adjacent tubulin (Fig. 4c and Extended Data Fig. 6b, 6c), driving the concerted compaction of the lattice. Aromatic π - π stacking interactions are known to form mechanical relays in proteins for transmission of conformation changes⁴⁸, including notably for hemoglobin, where flipping of a phenylalanine facilitates its transition between the relaxed and tense states⁴⁹. Aromatic relays have also been employed in protein engineering to propagate signals in the nanoscale⁵⁰. Notably, F202 of the Phe pair also connects through a network of hydrophobic interactions to helix H12 (Extended Data Fig. 6d), which is a binding platform for motors and MAPs that could therefore allosterically regulate lattice spacing through this network.”

We would predict that a mutation in the Phe pair would affect the lattice expansion, but our attempt at making a Phe255 to Ala mutation did not result in good enough expression for biophysical and nucleotide analysis. The next step is to refine expression and purification and do a systematic mutation to other amino acids to see whether we can find a variant that is suitable for biophysical analysis. As this reviewer knows, recombinant tubulin expression and purification takes time and is difficult, which is why it has been a stumbling block in the field for many years. Going through all these mutations and protocol optimization is a worthwhile future endeavor, but beyond the scope of this manuscript which does not focus on GTP hydrolysis conformational switching, but on revealing the first atomic resolution structure of a microtubule in a living axon, and the conformational state of the tubulin in these microtubules.

Minor concerns:

1. The author should provide the Q-value, which shows the resolvability of the key residues mentioned in their analysis.

Thank you for this comment. We now provide the Q values for all residues in a new table as Supplementary Table 2. As can be seen Q values for residues included in detailed analysis are 0.5-0.7, consistent with our overall 2.7Å resolution and the high quality of our map. We have also now included new panels that show the maps in the regions discussed. These show clear definition for side chains (Extended Data Figure 6). We also deposited all maps.

“2. In the introduction (Line 92), the authors stated that tubulin PTM enzymes are sensitive to lattice spacing, and the PTM status regulates how other proteins interact with microtubules and microtubule growth dynamics.

This statement is inaccurate, as only VASH1/SVBP (the microtubule detyrosinase) binding is shown to be sensitive to the conformational state of tubulin in the lattice. It is misleading to include the properties not associated with microtubule detyrosination/tyrosination (i.e., microtubule severing enzymes) into this sentence.”

Thank you for the careful reading. Yes, VASH1/SVBP is sensitive to lattice spacing, and more recently it has been reported that MATCAP1 also prefers the expanded lattice. The microtubule severing enzyme katanin is sensitive to the tyrosination/detyrosination status of the microtubule, showing higher activity on tyrosinated microtubules (Szczesna et al, Dev Cell 2022). We now state clearly it is katanin, as opposed to the generic microtubule severing enzyme term we used in the original submission. We now state on page 4:

“The microtubule detyrosination status, in turn, regulates kinesin and dynein motors²⁸⁻³¹, which carry diverse cargo, the microtubule severing enzyme katanin³², which regulates microtubule density and organization, and microtubule end-binding protein complexes, which regulate microtubule growth dynamics^{33,34}.”

Reviewer #3

In this manuscript, Zehr and colleagues used the human i3Neurosphere model to investigate the ultrastructure of axonal microtubules using cryo-EM and cryo-FIB milling. These techniques allowed them to resolve the native-state structure of tubulin in neuronal axons at atomic resolution. The authors found that axonal MTs in the human i3Neurosphere exhibit an expanded lattice structure, resembling that of GTP-bound MTs, even though the exchangeable site on β -tubulin was actually bound by GDP. Furthermore, they demonstrated that the inter-dimer interface of GDP-bound tubulin in axonal MTs closely resembles that of GTP-bound tubulin, suggesting that lattice expansion occurs despite GDP occupancy. They find that this expanded lattice conformation is associated with neuronal differentiation.

This manuscript aligns well with the growing interest of applying cryo-ET and cryo-EM in molecular and cellular neurobiology. Importantly, their study offers detailed mechanistic insights at the molecular level into the arrangement of tubulins within the MT lattice in neuronal axons. These findings contribute to a deeper understanding of how MTs support axon formation. While there is no doubt that this work is conceptual significant and is done at a very high technological level, there are several critical points that need to be addressed in the revised manuscript.

We thank the reviewer for their strong support of the significance and quality of our study.

Major comments:

1. The approach used to study axonal microtubules in this work closely resembles that of Hoffmann et al. (2021, DOI: 10.7554/eLife.70269). In their study, human brain organoids were used instead of iPSCs to generate axon bundles that grew directly on EM grids, followed by plunge freezing for structural analysis. Although the resolution achieved by Hoffmann et al. was lower, they had already resolved the ultrastructure of axonal MTs in human neurons. This prior work is not cited in the submitted manuscript.

Thank you for this. We have now added this citation in our revised introduction on pages 6-7. As the reviewer states, the Hoffman *et al* study did not provide molecular information about microtubules and allowed only a description of overall microtubule arrangement in the axon; however, it did show the potential of the neurosphere sample configuration in identifying and imaging axons using cryo-ET. We now state in our revised manuscript:

“This radial configuration allows easy identification of axons, and, critically, areas with thin ice, enabling single-particle data collection from hundreds of axons for high-resolution structural determination. An analogous experimental configuration was previously used with human cerebral organoids, but was limited to cryo-ET ultrastructural characterization, at nm resolution⁴² “

2. In line 111 of the manuscript, the authors claim that their work reveal “the first atomic resolution structure of microtubule that effectors generate and experience in vivo in the axon”. However, their whole experimental set up is based on cultured iPSCs. So, their work is purely an in vitro cell culture study. The authors should also state it clearly.

The structure we present is in a living axon. To avoid confusion between *in vivo*, in an animal, versus *in vivo*, in cell culture, we have made sure to change *in vivo* to “*in cellulo*” or “in a cell” or “in a living axon” throughout the text.

3. In the section of “Lattice expansion accompanies neuronal differentiation”, the authors compared MT lattice in the axon of differentiated neurons to undifferentiated iPSCs and stated that the expansion of MT lattice in neuronal axon is part of the neuronal differentiation program (line 239-240 and figure 4). To my opinion, the evidence are incomplete. Their data certainly indicate that the undifferentiated iPSCs has more compact MT lattice than the axon of differentiated neurons. However, it is possible that this expansion of MT lattice is taking place

globally in differentiated i3Neurons. To clarify this, the authors should also analyze MT lattice of somatodendritic area of the differentiated i3Neurosphere.

We thank the reviewer for this comment. We agree that the general reader would benefit from a more explicit discussion of the technological advance in imaging at high-resolution large number of axons, as well as the potential limitations of the approach we introduce, and therefore we have rewritten the results section to better cover this. This revised section starts on page 6. Our work reveals an expanded microtubule lattice in the axon, which is a novel and unexpected discovery, as the reviewers state. It is possible that, as part of neurogenesis, microtubules also expand their lattices in other compartments. The procedure described in our manuscript is not suitable for lattice analysis in the somatodendritic compartment. We state this now clearly at the top of page 7. We seed neurospheres, which contain cell bodies and proximal processes, at the edge of EM grids (Extended Data Figure 1). This positioning is critical because TEM requires thin vitreous ice. To prepare lamellae near the neurosphere core, neurospheres would instead need to be seeded closer to the grid center. However, due to the thickness of the neurosphere, blotting in this configuration would not yield sufficiently thin ice for electron penetration. Instead, a dome of thick ice would form over the neurosphere. Such grids are not amenable to EM screening, and targeting regions of interest for FIB milling would be impossible. As a result, neither grid inspection nor lamella preparation would be feasible in the somatodendritic region. An alternative strategy would be required to isolate somatodendritic microtubules. This could potentially be achieved by culturing dissociated neurons directly on EM grids at low density. However, these areas are not as thin as the thin axons where we collected our data and will require milling. Milling close to the cell body, however, is substantially more challenging due to geometric constraints: (i) the soma's apical–basal height (z-axis) is five to ten times greater than that of a neurite. Capturing these compartments in a lamella is therefore not trivial, and (ii) lamellae prepared at somatodendritic regions are typically triangular, with the soma forming the wide base and the neurite forming the narrow tip. The tip frequently breaks due to its fragility. Importantly, lamellae would inevitably contain microtubules from a mixture of cell bodies, axons, and dendrites, which cannot be reliably distinguished. The orientation of microtubules in these areas would also be random, further

complicating analysis as lattice spacing analysis requires good imaging of a minimal microtubule length.

The advantage of our iPSC-derived neurosphere-based approach is that it enables reliable identification of hundreds of axons for high-resolution structural characterization. Confident identification of the process type in a dissociated culture on the EM grid would require a fluorescent compartment-specific marker, which we would have to generate, validate in our i3Neurons, and establish it works well with a new cryo-EM workflow. While this would in principle allow milling at somatodendritic sites and the generation of a statistically meaningful dataset, it would entail developing and troubleshooting a whole new sample preparation and imaging pipeline, which we hope the reviewer will agree would be well beyond the scope of this manuscript. We now clearly indicate on page 6 of the manuscript that our sample preparation and data collection/analysis allows data collection only in the axonal compartment:

“Somatodendritic processes are not amenable to transmission electron microscopy (TEM) analysis in this configuration due to specimen thickness close to the neurosphere core, which is formed by cell bodies and proximal processes in thick vitreous ice, resulting in low-quality images from mixed cellular compartments.”

and we now state in Discussion:

“We note that while our work reveals a lattice conformational switch in the axon, it also remains possible that such a switch is operational in somatodendritic processes, which also undergo microtubule stabilization during differentiation. This will be the focus of future studies, together with a detailed analysis of the time dependence of this lattice conformational switch.”

However, we would like to emphasize that our analysis is the first of its kind. In this manuscript (1) we show that atomic resolution is achievable for microtubule reconstructions in a living cell,

allowing detailed visualization of the active site and even ordered water molecules, and therefore facilitating a mechanistic understanding of microtubule conformation in a human neuron; (2) we report the first 3D structure for an expanded microtubule lattice in cells providing evidence for such a conformation, and we show that this expanded conformation is acquired in the axon as part of neurogenesis, and (3) we present a robust pipeline for collecting and processing atomic resolution data of microtubules in the axons of i3N iPSC derived neurons. i3Ns offer highly reproducible, stereotyped neuron generation and ease of genetic engineering, opening the door to atomic-resolution mechanistic dissection of the effects of diverse microtubule regulators in the human axon. Furthermore, the availability as part of the iNDI initiative (iPSC for neurodegeneration disease initiative) of hundreds of i3 lines with mutations found in Alzheimer's, FTD, Parkinsons and other neurodegenerative diseases with cytoskeleton involvement promises to transform the analysis of microtubule defects in these disorders when combined with the structural analyses that we now show are possible in our study.

4. In the introduction section, the authors stated that Tau, which is an important protein involved in several neurological diseases, preferably binds to compact MTs and is also compacting MT lattice (line 91 & 92). The authors should also briefly discuss how could Tau still bind to MTs, if the axonal MT lattice are in an expanded state which is not preferred by Tau.

The reviewer brings up an interesting point. Tau prefers the compacted lattice, but it can still bind with high affinity (above its cellular expression level) to the expanded GMPCPP lattice also. Interestingly, *in vitro* reconstitution studies showed that tau forms dense assemblies on GDP microtubules, but not on expanded GMPCPP lattices. We have now added a paragraph in the discussion to cover this topic. We state:

“In vitro reconstitution studies showed that MAP tau, which is highly abundant in the axon, assembles into cohesive envelopes on brain GDP microtubules, but not on expanded GMPCPP microtubules¹⁷. The cohesive tau envelopes prevent kinesin-1 access to the microtubule. In contrast, kinesin-1 can access GMPCPP microtubules even if they are

covered with tau^{17,26}. It is interesting to speculate that the expanded conformation of the microtubule lattice in the axon might prevent tau from assembling into highly cohesive envelopes, therefore allowing selective access to motors. “

5. As already showed by previous literatures that the axon of neurons (as well as dendrite) have both dynamic and stable MTs. The authors should comment whether they could distinguish MTs with different dynamics in their tomography/single particle data set.

We cannot obtain information about dynamics from cryo-EM. The microtubules with different dynamics could potentially be distinguished through their posttranslational modifications, although the relationship between PTMs and dynamics is not always a straightforward one. Assigning posttranslational modifications to microtubules throughout the axon and coupling these with lattice measurements (which requires nm precision alignment given the close proximity of microtubules within the axonal bundle) will be a considerable and exciting technical challenge for the future.

Minor comments:

1. The authors should explain more clearly what is meant by the compact and expanded states of the MT lattice. Clarifying this would help make the manuscript more accessible to a broader readership.

Thank you for this suggestion. We have now added an explanatory sentence in the introduction on page 3 in the Introduction. We now state:

“In vitro structural studies of mammalian microtubules showed that GTP hydrolysis controls microtubule lattice conformation, with the stable GTP-lattice existing in an expanded state, and the unstable, GDP-lattice in a compacted state⁴⁻⁶. The expanded

microtubule lattice has a wider spacing between tubulin dimers, with a more open interface between them, while the compacted lattice has a tighter arrangement of tubulin dimers.”

2. Some of the abbreviations is not spelled out. For example, FSC in line 119 is one of the first abbreviation appears at the beginning of the results section, but it is not explained.

Thank you for the careful reading. We have corrected this throughout the manuscript.

3. It happens very often that the authors make a statement but does not reference the figure (e.g. from line 159 to 160, authors should also cite figure 1 at the end of the sentence).

We have addressed this throughout the manuscript.

4. There is no scale bar for figure 4a and b.

We added this to new Figure 5 (Figure 4 in the original submission). Thank you for the careful reading.

Reviewers 1 and 2 have no additional comments and support publication of the manuscript as is. We address below the comments from Reviewer 3.

Reviewer #1:

“The revised manuscript satisfactorily addresses my questions. This is a very nice study, and the findings will have a significant impact on the field.”

Reviewer #2:

“The authors have addressed the reviewers' questions/concerns. I support the publication of the manuscript in Nature Structural & Molecular Biology.”

Reviewer # 3:

“The authors have addressed most of my concerns in the revised manuscript and I appreciate the effort they have made in improving the manuscript. Without asking for more experiments, I think there are still a few points the authors can easily do to make their work more complete. These points are related to their reply to my major comment #3. I understand that only the axonal bundles are vitrified in their work flow, and the neurospheres would indeed be too thick to be vitrified properly via plunge freezing. But this can be achieved alternatively by high-pressure freezing and then combined with the lift-out method, which has been proven possible to prepare lamella from big chunk of materials. I think the authors could very briefly discuss the possibility of doing this in their iPSC-derived neurosphere to resolve MT lattice in the somatodendrite as their future perspective.”

We do not think the high pressure freezing and lift-out approach would work well for our purposes. The resulting tomograms would contain a mixture of cell bodies, axons and dendrites in different spatial views relative to the electron beam (e.g., end-on versus longitudinal views) that would be non-trivial to distinguish one from the other to assign the MT lattice parameters to a particular cell compartment. The lift-out approach as a technique is quite challenging and unlikely to generate sufficient data for meaningful statistical analyses within a reasonable timeframe, unless extensive instrument and microscope time are available.

“In their reply, the authors also mentioned that the somatodendritic area of cultured neurons would be too thick to be imaged via cryo-EM. However, this could be solved by imaging distal dendrites. If they would do this and look into MTs, the outcome would be sufficient to claim if the expansion of MT lattice happens globally in all neurites, or it is a specific thing only for axonal MTs. I am not asking for more experiments, but I suggest to discuss this in the manuscript.”

We mention this in the main text at the top of page 7. We note that it would not be easy to distinguish between dendrites and axons in this configuration and this assignment would have to rely on determining microtubule polarity or by creating reporter lines that can selectively mark only dendrites or axons.